# SIGNATURE KERNEL CONDITIONAL INDEPENDENCE TESTS IN CAUSAL DISCOVERY FOR STOCHASTIC PROCESSES

**Georg Manten**\* **& Cecilia Casolo**\*
Technical University of Munich
Helmholtz Munich
Munich Center for Machine Learning
{georg.manten, cecilia.casolo}@tum.de

**Emilio Ferrucci**
Mathematical Institute
University of Oxford
emilio.rossiferrucci@maths.ox.ac.uk

**Søren Wengel Mogensen**
Department of Automatic Control
Lund University
swm.fi@cbs.dk

**Cristopher Salvi**
Department of Mathematics
Imperial College London
c.salvi@imperial.ac.uk

**Niki Kilbertus**
Technical University of Munich
Helmholtz Munich
Munich Center for Machine Learning
niki.kilbertus@tum.de

## ABSTRACT

Inferring the causal structure underlying stochastic dynamical systems from observational data holds great promise in domains ranging from science and health to finance. Such processes can often be accurately modeled via stochastic differential equations (SDEs), which naturally imply causal relationships via 'which variables enter the differential of which other variables'. In this paper, we develop conditional independence (CI) constraints on coordinate processes over selected intervals that are Markov with respect to the acyclic dependence graph (allowing self-loops) induced by a general SDE model. We then provide a sound and complete causal discovery algorithm, capable of handling both fully and partially observed data, and uniquely recovering the underlying or induced ancestral graph by exploiting time directionality assuming a CI oracle. Finally, to make our algorithm practically usable, we also propose a flexible, consistent signature kernel-based CI test to infer these constraints from data. We extensively benchmark the CI test in isolation and as part of our causal discovery algorithms, outperforming existing approaches in SDE models and beyond.

## 1 INTRODUCTION

Understanding cause-effect relationships from observational data can help identify causal drivers for disease progression in longitudinal data and aid the development of new treatments, act upon the underlying influences of stock prices to support lucrative trading strategies, or speed up scientific discovery by uncovering interactions in complex biological systems such as gene regulatory pathways. Causal discovery (or causal structure learning) has received continued attention from the scientific community for at least two decades (Glymour et al., 2019; Spirtes et al., 2000; Vowels et al., 2022) with a notable uptick in previous years particularly regarding differentiable score-based methods (Zheng et al., 2018; Brouillard et al., 2020; Charpentier et al., 2022; Hägele et al., 2023; Lorch et al., 2021; Annadani et al., 2023; Zheng et al., 2020), building on score matching algorithms (Rolland

---

\*Equal contribution

et al., 2022; Montagna et al., 2023b;a), and other deep-learning based approaches (Chen et al., 2022; Ke et al., 2023; Yu et al., 2019; Ke et al., 2020). Many of these approaches aim at improving the scalability of causal discovery in the number of variables and observations as well as at incorporating uncertainty or efficiently making use of interventional data.

However, causal discovery from time series data has received much less attention and been mostly neglected in these recent advances. At a fundamental level, causal effects in dynamical systems can only 'point into the future', which should make causal discovery in time resolved data intuitively simpler. Nevertheless, except for restricted settings, this promise has not been realized methodologically and causal discovery in time series, especially for systems evolving in continuous time, remains a major challenge (Singer, 1992; Runge et al., 2019; Lawrence et al., 2021; Runge et al., 2023).

**Data generating process.** We assume data to follow a stochastic process $X := (X^1, \ldots, X^d)$, with $X^k$ taking values in $\mathbb{R}^{n_k}$ ($n_k \geq 1$), and $X$ satisfying the following system of stationary, path-dependent SDEs

$$\begin{cases} \mathrm{d}X_t^k = \mu^k(X_{[0,t]})\mathrm{d}t + \sigma^k(X_{[0,t]})\mathrm{d}W_t^k, \\ X_0^k = x_0^k \qquad \text{for } k \in [d] := \{1, \ldots, d\}. \end{cases} \tag{1}$$

We call this the *SDE model*. The subscript $[0,t]$ at $X$ means that $\mu^k$ (the 'drift') and $\sigma^k$ (the 'diffusion') are functions of the entire solution up to time $t$, i.e., they are defined on $C([0, +\infty), \mathbb{R}^n)$ (or some suitable subspace thereof), where $n := n_1 + \ldots + n_d$. Each $W^k$ is an $m_k$-dimensional Brownian motion, and $\sigma^k$ maps to $n_k \times m_k$-dimensional matrices. The noises $W^k$ together with the (possibly random) initial conditions $x_0^k$ are jointly independent. Therefore, the diffusion $\sigma$ of the entire system is block-diagonal (see equation 8). We refer to Rogers & Williams (2000) for details, Evans (2006) for a concise introduction, and Appendix A.2 for more intuition and discussion of the assumptions made here. Hence, the SDE together with a distribution over initial conditions defines our data generating process. Individual observations are paths, which in practice translate to stochastic, potentially irregularly sampled time series observations for $X_{[0,T]}^k$ with a maximum observation time $T$, see Figure 1. Appendix A.1 contains an overview table of the notations used.

**Induced causal graph.** Eq. (1) naturally implies cause-effect relationships: we call $i \in [d]$ a parent of $j \in V := [d]$ ($i \in \mathrm{pa}_j^{\mathcal{G}}$) when either $\mu^j$ or $\sigma^j$ is not constant in the $i$-th argument. These parental relationships define a directed graph $\mathcal{G} = (V, E)$, which we call *dependence graph* of the SDE model. The **goal of causal discovery** is to infer the graph $\mathcal{G}$ induced by the SDE model from a sample of observed solution paths, depicted in Figure 1. Here, we only consider data generating processes leading to directed graphs without cycles of length greater than one (i.e., only 'loops' $X^k \to X^k$ are allowed). We still call these directed acyclic graphs (DAGs) for simplicity.

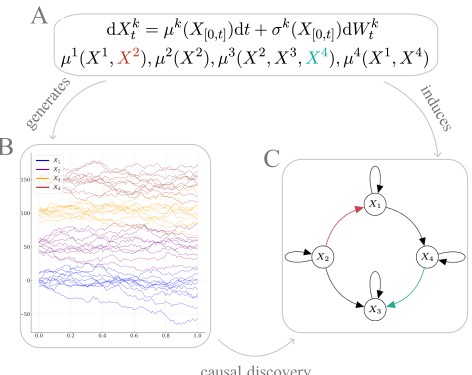

Figure 1: Illustration of causal discovery in the SDE model (A), leveraging conditional independencies in the observed samples (B) to infer the dependence graph of the SDE (C).

**Limitations and requirements.** The key limitation of our setting is the assumption of acyclicity (except for self-loops). We discuss fundamental issues arising from dropping the acyclicity constraint in Appendix A.3. Further, we focus on stationary SDE models, i.e., the coefficients in eq. (1) are not explicitly time-dependent, motivated by the requirement that causal relationships do not change over time. Despite these limitations, our setting is the first to simultaneously satisfy the following criteria for causal discovery in continuous time systems: (a) It does not rely on the 'discrete-time' assumption, i.e., that all variables are observed at the same, typically homogeneously spaced, time points (neither in the underlying model nor the actual observations). Instead, different variables can be observed at different, irregularly spaced time points in our setting. (b) We can handle partially observed systems with unobserved confounders. (c) Path-dependence ($\mu^k, \sigma^k$ may depend on the entire previous history $X_{[0,t]}$ of paths), including delayed SDEs, is typically not captured in existing approaches (Peters et al., 2022). (d) Beyond 'additive observation noise' it incorporates 'driving noise', where causal dependence may stem from the diffusion, not captured by the means of observed paths.

**Contributions.** Our main contributions include (a) developing CI constraints on coordinate processes over selected intervals that we prove to be Markov with respect to the acyclic dependence graph (with self-loops) induced by a general SDE model. (b) We then propose an efficient constraint based causal discovery algorithm and prove it to be sound and complete, assuming a CI-oracle, for both fully and partially observed data. Unlike existing constraint based methods in static settings, our algorithm uniquely recovers the full underlying graph or induced ancestral graph by exploiting the direction of time. (c) Finally, we propose a flexible, practical CI test on path space, rendering our method highly practicable. (d) We extensively demonstrate the test's efficacy and achieve superior performance in causal discovery compared to existing methods.

## 2 BACKGROUND AND RELATED WORK

**Causal discovery on time series.** While virtually all causal discovery methods on time series exploit that the direction of time constrains edges to point forward, except for Laumann et al. (2023), most existing work assumes observations to be sampled on a regular time grid with a discrete (auto-regressive) law, where past observations $(X_{t-\tau}, \ldots, X_{t-1})$ determine the present $X_t = f(X_{t-\tau}, \ldots, X_{t-1}, \varepsilon_t)$ for some fixed lag $\tau$, which can be seen as a static structural causal model (SCM) over an 'unrolled graph' with variables at different time steps considered as distinct nodes (Assaad et al., 2022; Hasan et al., 2023; Peters et al., 2017). Based on the seminal work (Granger, 1969), various Granger-type approaches leverage the assumption that past values of a variable can help predict future values to detect lagged causal influences both for linear (Diks & Panchenko, 2006; Granger, 1980) and non-linear (Marinazzo et al., 2008; Shojaie & Fox, 2022; Runge, 2020; Pamfil et al., 2020) functional relationships. While non-parametric constraint-based methods for such discrete-time models (Runge et al., 2019; 2020; 2023) can handle partial observations, they are also fundamentally limited by the 'discrete-time' assumption making them unfit for irregularly sampled observations of continuous time systems, path-dependence, or diffusion dependence. Furthermore, they require numerous tests across all time points and rely on the correct estimation of a fixed 'lookback window' $\tau$. The fundamental problems of the 'discrete-time' assumption have also been discussed in Runge (2018).

Other work focuses on the (static) equilibrium behavior of differential equations (Mooij et al., 2013; Bongers et al., 2018; Bongers & Mooij, 2018), exploiting invariance in mass-action kinetics (Peters et al., 2022), or attempts to directly learn the full dynamical law via non-convex optimization of heavily overparameterized neural network models (Aliee et al., 2021; 2022; Bellot et al., 2022; Wang et al., 2024). The latter are fundamentally limited by the fixed functional modeling structures and therefore also unable to model partially observed settings. Albeit its limited applicability to fully-observed, Markovian SDE models, SCOTCH (Wang et al., 2024), which uses a variational formulation to infer posterior distributions over possible graphs, is the current state-of-the-art baseline.

Constraint-based methods for continuous-time stochastic processes mostly rely on (conditional) local independence, an asymmetric independence relation (Schweder, 1970; Mogensen et al., 2018), which can be used to infer a (partial) causal graph in, e.g., point process models (Didelez, 2008; Meek, 2014; Mogensen et al., 2018; Mogensen & Hansen, 2020). However, besides the limitation to drift-dependencies, there exists no practical test of local independence for diffusions. The only work that actually tests local independence from data is (Christgau et al., 2023), which is heavily restricted to counting processes.

In this work, we overcome the challenges of irregular and partial observations, path- and diffusion dependence, and practical applicability by transforming CI statements of the form

$$X_{\mathcal{I}}^I \perp\!\!\!\perp X_{\mathcal{J}}^J \mid X_{\mathcal{K}}^K, \tag{2}$$

for disjoint $I, J, K \subseteq [d]$ between subsets of coordinate-processes $X^I$, $X^J$, $X^K$ over intervals $\mathcal{I}, \mathcal{J}, \mathcal{K} \subseteq [0, T]$ into a practical hypothesis test using the signature kernel and the arrow of time.

**Conditional independence tests.** As a key building block of graphical models, there is a large literature on conditional independence tests, i.e., testing the null hypothesis $H_0 : X \perp\!\!\!\perp Y \mid Z$. When $Z$ is discrete, CI testing reduces to a series of unconditional tests for each value of $Z$ (Tsamardinos & Borboudakis, 2010). For continuous $Z$, nonparametric kernel-based methods often either check independence of residuals after kernel (ridge) regressing $X$ and $Y$ on $Z$ (Shah & Peters, 2020; Lundborg et al., 2022; Zhang et al., 2011; Daudin, 1980; Strobl et al., 2019), or measure the distance

between joint and marginal kernel mean embeddings (Muandet et al., 2017; Park & Muandet, 2020). For example, KCIPT (Doran et al., 2014) phrases it as a two-sample test of the null $H_0$ : $P(X, Y, Z) = P(X \mid Z)P(Y \mid Z)P(Z)$ and simulates samples from $P(X \mid Z)P(Y \mid Z)P(Z)$ via a permutation of the data that approximately preserves $Z$. Lee & Honavar (2017) improve on KCIPT with an unbiased estimate of Maximum Mean Discrepancy (MMD) (Gretton et al., 2006). Other permutation-based approaches require large sample sizes (Sen et al., 2017) or rely on densities (Kim et al., 2022), unfit for our setting.

Laumann et al. (2023) develop a CI test for functional data using the Hilbert-Schmidt conditional independence criterion (HSCIC) (Park & Muandet, 2020) and a permutation test (Berrett et al., 2019), which they use as part of the PC-algorithm (Glymour et al., 2019) for causal discovery. Even though their approach assumes $P_{X|Z}$ to be known (not the case in our setting), it is inapplicable under partial observations, and does not exploit the arrow of time (to identify graphs beyond the Markov equivalence class), we benchmark the work by Laumann et al. (2023) against our method in the experiments. CI testing faces fundamental practical challenges, such as the exponential growth in the number of values to condition on as dimensionality of $Z$ increases, and theoretical limits, as any CI test's power is bounded by its size (Shah & Peters, 2020; Lundborg et al., 2022). This 'no free lunch' statement underscores the need for carefully selecting the right test. Most existing approaches assume Euclidean spaces and do not generalize to path-valued random variables without densities, a gap we fill by combining kernel-based permutation tests (KCIPT, SDCIT) with the signature kernel for a tailored CI test on path space and proving its consistency.

**Signature kernels.** Signature kernels (Király & Oberhauser, 2019; Salvi et al., 2021a), a universal class for sequential data, have received attention recently for their efficiency in handling path-dependent problems (Lemercier et al., 2021; Salvi et al., 2021c; Cochrane et al., 2021; Salvi et al., 2021b; Cirone et al., 2023; Issa et al., 2023; Pannier & Salvi, 2024). The definition of the signature kernel requires an initial algebraic setup, which we keep as concise as possible—yet self-contained— here and refer the reader to Cass & Salvi (2024, Chapter 2) for more details. We provide some more informal intuition of signatures in Appendix A.4. Let $\langle \cdot, \cdot \rangle_1$ be the Euclidean inner product on $\mathbb{R}^d$. Denote by $\otimes$ the standard outer product of vector spaces. For any $n \in \mathbb{N}$, we denote by $\langle \cdot, \cdot \rangle_n$ on $(\mathbb{R}^d)^{\otimes n}$ the canonical Hilbert-Schmidt inner product defined for any $a = (a_1, \ldots, a_n)$ and $b = (b_1, \ldots, b_n)$ in $(\mathbb{R}^d)^{\otimes n}$ as $\langle a, b \rangle_n = \prod_{i=1}^{n} \langle a_i, b_i \rangle_1$. Define the direct sum of vector spaces $T(\mathbb{R}^d) := \bigoplus_{n=1}^{\infty} (\mathbb{R}^d)^{\otimes n}$, where it is understood that the direct sum runs over finitely many non-zero levels. The inner product $\langle \cdot, \cdot \rangle_n$ on $(\mathbb{R}^d)^{\otimes n}$ can then be extended by linearity to an inner product $\langle \cdot, \cdot \rangle$ on $T(\mathbb{R}^d)$ defined for any $a = (a_0, a_1, \ldots)$ and $b = (b_0, b_1, \ldots)$ in $T(\mathbb{R}^d)$ as $\langle a, b \rangle = \sum_{n=0}^{\infty} \langle a_n, b_n \rangle_n$. Other choices of linear extensions have been studied in Cass et al. (2023). We denote by $\mathcal{T}(\mathbb{R}^d)$ the Hilbert space obtained by completing $T(\mathbb{R}^d)$ with respect to $\langle \cdot, \cdot \rangle$.

The *signature transform* is a classical path-transform from stochastic analysis. For any sub-interval $[s, t] \subset [0, T]$ and any continuous path $X \in C_p([0, T], \mathbb{R}^d)$ of finite $p$-variation, with $1 \leq p < 3$, it is (canonically) defined as $S(X)_{s,t} := (1, S(X)_{s,t}^{(1)}, \ldots, S(X)_{s,t}^{(n)}, \ldots) \in \mathcal{T}(\mathbb{R}^d)$, where $S(X)_{s,t}^{(n)} \in (\mathbb{R}^d)^{\otimes n}$ is the $n$-fold iterated integral $S(X)_{s,t}^{(n)} = \int_{s<u_1<\ldots<u_n<t} dX_{u_1} \otimes dX_{u_2} \otimes \ldots \otimes dX_{u_n}$. Given two arbitrary sub-intervals $[a, b], [c, d] \subset [0, T]$, the *signature kernel* $K_S : C_p([s, t], \mathbb{R}^d) \times C_p([s', t'], \mathbb{R}^d) \to \mathbb{R}$ is a positive definite kernel on continuous paths of bounded variation defined as

$$K_S(X, Y) = \langle S(X)_{s,t}, S(Y)_{s',t'} \rangle . \tag{3}$$

Salvi et al. (2021a, Thm. 2.5) shows that $K_S(X, Y) = f(t, t')$, where $f : [s, t] \times [s', t'] \to \mathbb{R}$ is the solution of the following path-dependent integral equation:

$$f(t, t') = 1 + \int_s^t \int_{s'}^{t'} f(u, v) \langle dX_u, dY_v \rangle_1 , \quad \text{with } f(0, \cdot) = f(\cdot, 0) = 1 . \tag{4}$$

This 'kernel trick' allows us to evaluate the signature kernel without explicit computation of the signature transform by solving the partial differential equation (PDE) in eq. (4). We refer to Salvi et al. (2021a) for a numerical approximation scheme to solve this hyperbolic PDE and its error rates and to Appendix A.4 for more mathematical details on the applicability in our setting. In our experiments, we use the JAX library `sigkerax` to efficiently solve eq. (4).

In summary, the signature kernel is a universal kernel that takes (multi-variate) continuous paths— potentially on different intervals—as input, is efficiently computable, and performs well in capturing characteristics of sequential data (Lee & Oberhauser, 2023).

## 3 METHODOLOGY

We start with our constraint-based causal discovery algorithms, where we assume an oracle for CI testing on path space for eq. (2) in Section 3.1. In Section 3.2 we then introduce the signature kernel CI test and state its consistency, which requires a new proof that does not rely on densities. Our test, applicable to arbitrary stochastic processes, time series, or functional data, performs well empirically in our extensive experiments in Section 4 making it a contribution of independent interest.

### 3.1 CAUSAL DISCOVERY IN THE ACYCLIC SDE MODEL

Throughout this section, we assume access to a CI oracle for eq. (2), where $X_{[a,b]}^H$ denotes the $C([a,b], \mathbb{R}^{|H|})$-valued random variable $\omega \mapsto ([a,b] \ni t \mapsto (X_t^h(\omega) - X_a^h(\omega))^{h \in H})$. Asymptotically, such an oracle can be replaced in practice by a consistent finite-sample CI test for path-valued random variables (see Section 3.2). More specifically, we test the independence of increments, not the independence of the processes on consecutive intervals, meaning that we consider path-valued random variables $\omega \mapsto ([a,b] \ni t \mapsto (X_t^h(\omega) - X_a^h(\omega))^{h \in H})$, effectively decoupling initial conditions from subsequent increments. For simplicity and since this minor subtlety is naturally handled by the signature kernel as the signature transform is translation invariant (meaning $S(X(t))_{a,b} = S(X(t) - X(a))_{a,b}$, we denote these paths by $X_t^{h \in H}$.

In Appendix A.3, we prove that when allowing for cycles in the SDE model, constraint-based causal discovery of the full graph is impossible using arbitrary expressions of the form in eq. (2) despite the flexibility of these conditional independencies. This impossibility result is a key motivation to study the acyclic setting still allowing for loops, which are crucial in dynamic settings as variables should be allowed to depend on themselves infinitesimally into the past.

We will use the oracle in the following ways for intervals $[0,s]$ and $[s, s+h]$ with $h > 0$:

- $X^i$ is *symmetrically CI* of $X^j$ given $X^K$ on $[0,T]$ if $X_{[0,T]}^i \perp\!\!\!\perp X_{[0,T]}^j \mid X_{[0,T]}^K$;
  we then write $X^i \perp\!\!\!\perp_{\text{sym}} X^j \mid X^K$

- $X^i$ is *future-extended h-locally CI* of $X^j$ given $X^K$ at $s$ if $X_{[0,s]}^i \perp\!\!\!\perp X_{[s,s+h]}^j \mid X_{[0,s]}^j, X_{[0,s+h]}^K$;[1]
  we then write $X^i \perp\!\!\!\perp_{s,h}^+ X^j \mid X^K$

- $X^i$ is *conditionally h-locally self-independent* given $X^K$ at $s$ if $X_{[0,s]}^i \perp\!\!\!\perp X_{[s,s+h]}^i \mid X_{[0,s+h]}^K$;
  we then write $X^i \perp\!\!\!\perp_{s,h}^{\circlearrowleft} \mid X^K$

The key idea to leverage the unidirectional flow of time is to split the observed paths into 'past' and 'future' at some time $t \in [0,T]$. Therefore, we define the *lifted dependence graph* $\tilde{\mathcal{G}} = (\tilde{V}, \tilde{E})$ by setting $\tilde{V} := V_0 \sqcup V_1$, where $\sqcup$ denotes disjoint union and $V_0, V_1$ are two copies of $V$, whose elements we subscript with 0 and 1; we include an edge $(i_0 \to i_1) \in \tilde{E}$ if and only if there is a loop $(i \to i) \in E$, and for each edge $(i \to j) \in E$ with $i \neq j$, include edges $(i_0 \to j_0), (i_1 \to j_1), (i_0 \to j_1) \in \tilde{E}$, see Figure 2. Conversely, we say that the graph with edges $E$ is obtained by *collapsing* the graph with edges $\tilde{E}$.
Intuitively, $V_0$ contains all 'past' and $V_1$ all 'future'

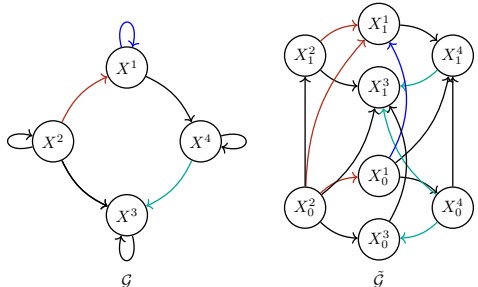

Figure 2: The lifted dependence graph $\tilde{\mathcal{G}}$ (right) for a DAG $\mathcal{G}$ (left) with colors highlighting the correspondence of selected edges.

variables with the same edges among them (causal relations) with additional edges from 'past' to 'future'. We can now establish the following Markov property.

**Proposition 3.1** (Markov property). *Assume the dependence graph $\mathcal{G}$ of the SDE model is acyclic except for loops and associate $X_{[0,s]}^i$ with $i_0 \in V_0$ and $X_{[s,s+h]}^i$ with $i_1 \in V_1$. Then $\tilde{\mathcal{G}}$ is acyclic and the independence relation $\perp\!\!\!\perp_{s,h}^+$ satisfies the global Markov property w.r.t. d-separation in $\tilde{\mathcal{G}}$.*

Our proof (see Appendix A.6) leverages the structure of the SDE-generating mechanism, which factorizes according to the dependence graph, thereby allowing us to formally establish the global

---

[1] The '+' denotes conditioning on the future of $X^K$, not common in the literature (Florens & Fougere, 1996).

Markov property using arguments from stochastic analysis. With an oracle for eq. (2) and a faithfulness type assumption (for more details see Appendix A.8), we now develop new constraint-based causal discovery algorithms that infer the full dependence graph instead of just equivalence classes.

**Discovering the full DAG $\mathcal{G}$.** To discover the full graph including loops we propose Algorithm 1 that makes use of the time-ordering via our $h$-local independence models (consider $s, h > 0$ arbitrary but fixed).

**Theorem 3.2.** *Algorithm 1 is sound and complete for the SDE model, assuming acyclicity except for loops and faithfulness: its output is the true dependence graph $\mathcal{G}$.*

The proof (see Appendix A.7) establishes completeness by identifying a separating set and leveraging the global Markov property in Proposition 3.1; soundness is shown via the faithfulness assumption. In particular, our proof does not require the 'strong faithfulness' assumption, but a weaker version, 'parent faithfulness', suffices—see Appendix A.8 for a detailed discussion of faithfulness in our setting.

---

**Algorithm 1: Causal discovery for acyclic SDEs.**

1: $\tilde{V} \leftarrow \{k_0, k_1 \mid k \in V\}$
   $\tilde{E} \leftarrow \{i_0 \rightarrow j_0,\ i_1 \rightarrow j_1 \mid i, j \in V,\ i \neq j\}$
   $\quad \cup \{i_0 \rightarrow j_1 \mid i, j \in V\}$
2: **for** $c = 0, \ldots, d-2$ **do**    ▷ *edge recovery w/o loops*
3: $\quad$ **for** $i, j \in V,\ i \neq j$ **do**
4: $\quad\quad$ **for** $K \subseteq V \setminus \{i, j\},\ |K| = c,$
        s.t. $(k_0 \rightarrow j_1) \in \tilde{E}$ for $k \in K$ **do**
5: $\quad\quad\quad$ **if** $X^i \perp\!\!\!\perp_{s,h}^+ X^j \mid X^K$ **then**
6: $\quad\quad\quad\quad$ $\tilde{E} \leftarrow \tilde{E} \setminus \{i_0 \rightarrow j_0, i_1 \rightarrow j_1, i_0 \rightarrow j_1\}$
7: $\mathcal{G} = (V, E) \leftarrow \text{collapse}(\tilde{V}, \tilde{E})$
8: **for** $k \in V$ **do**    ▷ *removing loops*
9: $\quad$ **if** $X^k \perp\!\!\!\perp_{s,h}^{\circlearrowleft} \mid X^{\text{pa}_k^{\mathcal{G}} \setminus \{k\}}$ **then**
10: $\quad\quad$ $E \leftarrow E \setminus \{k \rightarrow k\}$
11: **return** $\mathcal{G}$

---

**Discovering and post-processing the CPDAG.** In Algorithm 1, we leverage our Markov property with respect to the lifted graph that also incorporates time directionality to infer the full graph. We now show that we can also use the symmetric criterion $\perp\!\!\!\perp_{\text{sym}}$ to recover the 'Markov equivalence class'—all graphs that are Markov equivalent to $\mathcal{G}$—via the completed partially directed acyclic graph (CPDAG) of $\mathcal{G}$ (Peters et al., 2017, Def. 6.24). Loosely speaking, the CPDAG has the same adjacencies as $\mathcal{G}$, but some edges may remain undirected. Due to space limitations, we present the required global Markov property with respect to $\perp\!\!\!\perp_{\text{sym}}$ in Appendix A.9. This Markov property allows us to apply the sound and complete PC algorithm to infer the CPDAG assuming an CI oracle for $\perp\!\!\!\perp_{\text{sym}}$ and faithfulness (Spirtes et al., 2000). While Algorithm 1 is strictly more informative (returns $\mathcal{G}$ instead of just its CPDAG), the reliability of $\perp\!\!\!\perp_{s,h}^+$ in practice might be negatively affected compared to $\perp\!\!\!\perp_{\text{sym}}$ by (a) conditioning on larger sets, (b) shorter time segments (potentially losing information), and (c) the additional choice of parameters $s, h$. Hence, in real-world applications inferring the CPDAG using $\perp\!\!\!\perp_{\text{sym}}$ may be more robust than inferring $\mathcal{G}$ fully using $\perp\!\!\!\perp_{s,h}^+$. Building on these potential benefits, we also develop a post-processing procedure for the CPDAG that again leverages the directionality of time to provably also orient all remaining unoriented edges.

**Corollary 3.3** (post-processing). *Let $\mathcal{G} = (V, E)$ be the dependence graph of the acyclic (except for loops) SDE model and $\tilde{\mathcal{G}} = (V, \tilde{E})$ its CPDAG. Then we have $X_{[0,T]}^j \not\perp\!\!\!\perp X_0^i$ but $X_{[0,T]}^i \perp\!\!\!\perp X_0^j$ for all $(i, j) \in E \subset \tilde{E}$ with $(j, i) \in \tilde{E}$ and $i \neq j$.*

The proof in Appendix A.10 relies on the joint independence of Brownian motions $\mathrm{d}W_t^k$ and initial conditions $X_0^k$. Corollary 3.3 directly motivates an alternative algorithm for causal discovery of $\mathcal{G}$, written out as Algorithm 2 in Appendix A.10, by first constructing the CPDAG (PC algorithm with $\perp\!\!\!\perp_{\text{sym}}$) followed by testing unconditionally the yet unoriented edges as in Corollary 3.3.

**Partially observed setting.** Finally, we consider the partially observed setting, where only $\{X_t^{v_k}\}_{v_k \in V_{obs}, t \in [0,1]}$ of the process $\{X_t^{v_i}\}_{v_i \in V_{obs} \sqcup V_L, t \in [0,1]}$ from the model in eq. (1) with DAG $\mathcal{G} = (V_{obs} \sqcup V_L, E)$ are observed. Asymmetric local independence based methods for causal discovery are challenged by requiring exponentially many oracle calls in the partially observed setting (Mogensen & Hansen, 2022; Mogensen, 2025). However, since our symmetric conditional independence constraint satisfies the global Markov property with respect to the induced acyclic directed mixed graph (ADMG) $\mathcal{G}_{obs} = (V_{obs}, E')$ (also known as latent projection, with potentially arbitrary unobserved processes) over the observed variables, we can take inspiration from the Fast Causal Inference (FCI) algorithm (Zhang, 2008) by first running its skeleton-discovery-part and then again leveraging the direction of time to uniquely infer the edge-type $\{\leftarrow, \rightarrow, \leftrightarrow\}$ for found adjacencies. This reduces the typical partial ancestral graph output of FCI (representing an equivalence class of

ancestral graphs) to a single, maximally informative graph (as informative as an ancestral graph can be). We provide all details in Appendix D.

## 3.2 SIGNATURE KERNEL CONDITIONAL INDEPENDENCE TEST

Our theoretical results in Section 3.1 assume a CI oracle. To make our algorithms usable in practice, we now propose a flexible CI test for path-valued random variables on different time intervals for expressions of the form eq. (2). Recent work using the signature kernel is limited to unconditional hypothesis tests (Chevyrev & Oberhauser, 2022) or only use it as a heuristic measure of conditional independence (not developing a hypothesis test) (Salvi et al., 2021c), and both do not use the time order. Using terms of the signature to detect causality (not specifically with hypothesis testing) was also proposed by Giusti & Lee (2020); Glad & Woolf (2021). Hence, we are the first to combine ideas from kernel-based CI tests and the signature kernel for a practically usable, consistent CI test on path space. While the existing consistency proof for KCIPT (also applying to SDCIT) (Doran et al., 2014) relies on the existence of densities, we prove consistency for testing on path-valued random variables in Appendix A.14 requiring novel arguments. We discuss the sensitivity of constraint-based causal discovery on errors in the CI test in Appendix A.12. Given $n$ samples of path segments $X_{\mathcal{I}}^{(i)}, Y_{\mathcal{J}}^{(i)}, Z_{\mathcal{K}}^{(i)}$ on intervals $\mathcal{I}, \mathcal{J}, \mathcal{K} \subset [0, T]$ for $i \in [n]$, we compute the Gram matrices $k_{XX}, k_{YY}, k_{ZZ} \in \mathbb{R}^{n \times n}$ using the signature kernel $K_S$ and run a kernel-based permutation CI test like KCIPT (Doran et al., 2014) or SDCIT (Lee & Honavar, 2017) to test $X_{\mathcal{I}} \perp\!\!\!\perp Y_{\mathcal{J}} \mid Z_{\mathcal{K}}$.[2] The theoretical foundation combined with the strong empirical performance of our CI test suggest that it may be of independent interest outside of causal discovery applications.

## 4 EXPERIMENTS

In this section, after introducing the baselines and implementation details, we empirically evaluate our CI test in a variety of settings. We then demonstrate strong performance in various causal discovery tasks and also outperform traditional methods in a real-world finance case study.

**Baselines.** We compare against CCM (Sugihara et al., 2012), PCMCI (Runge et al., 2019), Granger causality (Granger, 1969), a kernel-based approach (Lau) (Laumann et al., 2023), and SCOTCH (Wang et al., 2024), a variational neural SDE method. CCM, Granger, and PCMCI are well-established methods for time series, while Lau and SCOTCH represent the latest advances in causal discovery for functional data and SDE models, respectively. CCM and Granger are limited to bivariate cases, while Lau is only applicable to unconditional tests here. Details are in Appendix C.

**Implementation details and metrics.** We use `sigkerax` for the signature kernel with an RBF kernel with length scale selected via a median heuristic (see Appendix B.1). For bivariate cases, the causal structure is $X^1 \to X^2$, for $d > 2$, we draw Erdös–Rényi DAGs. Performance is measured by Structural Hamming Distance (SHD, Appendix B.8). For $\perp\!\!\!\perp_{s,h}^+$, $s = 0.1 \cdot T$ (and a fixed $T = 1$) performed best (Table 5). The SDE model is

$$\mathrm{d}X_t = (AX_t + c)\mathrm{d}t + \mathrm{Diag}(BX_t + d)\mathrm{d}W_t, \quad \text{with } A, B \in \mathbb{R}^{d \times d}, c, d \in \mathbb{R}^d . \tag{5}$$

except for the 2-dimensional non-linear SDE, which is given by

$$\mathrm{d}\begin{pmatrix} X_t^1 \\ X_t^2 \end{pmatrix} = \begin{pmatrix} -r\omega \sin(\omega t) \\ r\omega \tanh(X_t^2) \end{pmatrix} \mathrm{d}t + \mathrm{Diag}(d^\top)\mathrm{d}W_t . \tag{6}$$

**Computational complexity.** Due to space constraints, we defer the computational complexity analysis of our CI test, the causal discovery algorithm, and the signature kernel evaluations in Appendix B.9. The overall computation is dominated by finding the permutation that leaves $Z$ (approximately) invariant in the permutation based CI tests (KCIPT, SDCIT).

**Choosing (conditional) independence tests.** In initial experiments on two- and three-dimensional structures (e.g., forks, colliders, chains) detailed in Appendix B.2, we found bootstrapped SDCIT and HSIC to be the best performing variants of kernel-based (C)I tests (see Figure 6 and Tables 6 to 8 for details) and use them for all subsequent experiments.

---

[2] While time observations within a process are not i.i.d., the $n$ process samples are i.i.d. realizations of the SDE model, see Appendix A.13 for details.

Table 1: SHD ($\times 10^2$) comparison of SigKer to the baselines in four bivariate SDE settings: linear, path-dependence, non-linear, and diffusion dependence. We ran SCOTCH for different sparsity parameters $\lambda$ and numbers of epochs $n_e$; there is no clear trend in either parameter, but $\lambda = 100, n_e = 2000$ performed best overall. Different learning rates performed worse across the board.

| $\times 10^2$ | $(\lambda, n_e)$ | linear | | path-dependence | | non-linear | | diffusion dependence | |
|---|---|---|---|---|---|---|---|---|---|
| | | $n = 200$ | $n = 400$ | $n = 200$ | $n = 400$ | $n = 200$ | $n = 400$ | $n = 200$ | $n = 400$ |
| CCM | | $176 \pm 7$ | $166 \pm 8$ | $186 \pm 5$ | $194 \pm 3$ | $120 \pm 10$ | $106 \pm 10$ | $100 \pm 10$ | $84 \pm 10$ |
| Granger | | $92 \pm 5$ | $87 \pm 5$ | $85 \pm 5$ | $95 \pm 6$ | $103 \pm 5$ | $103 \pm 5$ | $105 \pm 5$ | $111 \pm 4$ |
| PCMCI | | $89 \pm 9$ | $100 \pm 10$ | $91 \pm 4$ | $125 \pm 15$ | $41 \pm 8$ | $55 \pm 15$ | $98 \pm 13$ | $75 \pm 23$ |
| SCOTCH | $100, 1k$ | $74 \pm 14$ | $77 \pm 17$ | $50 \pm 12$ | $46 \pm 12$ | $64 \pm 15$ | $80 \pm 10$ | $75 \pm 14$ | $62 \pm 10$ |
| SCOTCH | $50, 2k$ | $79 \pm 18$ | $44 \pm 17$ | $23 \pm 12$ | $73 \pm 23$ | $83 \pm 11$ | $73 \pm 13$ | $\mathbf{25 \pm 13}$ | $44 \pm 28$ |
| SCOTCH | $100, 2k$ | $50 \pm 17$ | $36 \pm 19$ | $64 \pm 19$ | $55 \pm 15$ | $85 \pm 7$ | $91 \pm 9$ | $33 \pm 16$ | $\mathbf{9 \pm 8}$ |
| **SigKer** | | $\mathbf{14 \pm 4}$ | $\mathbf{7 \pm 3}$ | $\mathbf{5 \pm 2}$ | $\mathbf{6 \pm 2}$ | $\mathbf{28 \pm 5}$ | $\mathbf{5 \pm 2}$ | $72 \pm 6$ | $63 \pm 5$ |

**Power analysis of the unconditional test.** We first demonstrate consistently superior performance of our unconditional independence test (HSIC) SigKer (ours) over the only existing baseline for this setting from Laumann et al. (2023) (Lau) in the linear SDE-setting eq. (5) (with $B \equiv 0$, $d_i = 0.4$), reaching test power near 1 already for $n \geq 40$ when the causal interaction $a_{21}$ (strength of $X^1 \to X^2$) is comparable to $X^2$'s self-dependence ($a_{22}$) in Figure 3.

Figure 7 further highlights SigKer's robustness even with *high data missingness*, and Figure 8 demonstrates its effectiveness on *fractional Brownian motions* outside the semi-martingale framework. We are not aware of other CI tests that apply to such settings—establishing it as state-of-the-art CI test for stochastic processes.

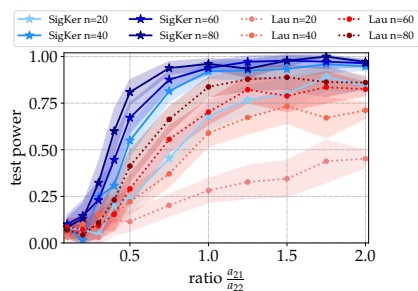

Figure 3: Test power for $X^1_{[0,t]} \perp\!\!\!\perp X^2_{[0,t]}$ over $\frac{a_{21}}{a_{22}}$. Lines (shades) are means (standard errors) over 1000 SDE instances.

**Leveraging the direction of time.** Next, we leverage time to orient the edge in the much-studied bivariate causal-discovery setting ($X^1 \to X^2$) via our $\perp\!\!\!\perp^+_{s,h}$ constraint with $K = \emptyset$ for a variety of settings: (i) linear SDEs with dependence in the drift, (ii) diffusion, (iii) or path-dependent, as well as (iv) nonlinear interactions. Table 1 shows that SigKer decisively dominates all baselines in all settings for different sample sizes except diffusion dependence. In this setting, SCOTCH is better *for specific hyperparameter settings*, while performing worse for others. Crucially, one cannot 'cross-validate' or otherwise select such hyperparameters for the causal-discovery task in a data-driven fashion without ground-truth knowledge. The lack of robustness (different optimal hyperparameters for different sample sizes) renders SCOTCH unreliable in practice even for diffusion dependence.

Specifically, we draw the settings for these experiments as follows: for linear drift interactions we sample $a_{21} \sim \mathcal{U}([1, 2.5])$, $a_{11}, a_{22} \sim \mathcal{U}([-0.5, 0.5])$, $a_{12} = 0$, $B \equiv 0$, $d_i \sim \mathcal{U}([0.1, 0.2])$ in a two-dimensional linear SDE model eq. (5). For linear diffusion interactions we draw $a_{11}, a_{22} \sim \mathcal{U}([0.5, 1])$ and $b_{21} \sim \mathcal{U}([1, 4.5])$), the rest set to zero in a two-dimensional linear SDE model eq. (5). For path-dependence, we simulate $dX^2_t = \mu^2(X^1_{[0,t]})dt + d_2 dW^1_t$ via a three-dimensional SDE eq. (5) with $B \equiv 0$, $c \equiv 0$, $a_{23}, a_{31} \sim \mathcal{U}([-3.5, -1] \cup [1, 3.5])$ and $d = (d_1, d_2, 0)^\top$, $d_i \sim \mathcal{U}([0.1, 0.2])$, the rest set to zero. For the nonlinear SDEs we use eq. (6) with $\omega \sim \mathcal{U}([6\pi, 8\pi])$, $r \sim \mathcal{U}([0.5, 1])$, $d_i \sim \mathcal{U}([2, 2.5])$.

**Causal discovery.** For causal discovery, we sample 40 linear SDEs from eq. (5) (with $a_{ij} \sim \mathcal{U}([-2, -1] \cup [1, 2])$ for $j \neq i$ and $a_{ii} \sim \mathcal{U}([-0.5, 0.5])$) with random DAG adjacency structures for $d \in \{3, 5, 10, 20, 50\}$ and use $n = 200$ sample paths from each SDE as input for the algorithms. Table 2 shows that

Table 2: SHD ($\times 10^2$) comparison of SigKer to PCMCI and SCOTCH (different $\lambda$ and $n_e$) in causal discovery. Means and standard errors are over 40 SDE instances.

| | $d = 3$ | $d = 5$ | $d = 10$ | $d = 20$ | $d = 50$ |
|---|---|---|---|---|---|
| PCMCI | $29 \pm 16$ | $243 \pm 37$ | $793 \pm 84$ | $3530 \pm 159$ | $19.6k \pm 857$ |
| SCOTCH 100, 2k | $188 \pm 28$ | $417 \pm 86$ | $250 \pm 61$ | $1525 \pm 1160$ | $10275 \pm 6176$ |
| SCOTCH 200, 2k | $110 \pm 21$ | $270 \pm 48$ | $530 \pm 223$ | $\mathbf{370 \pm 174}$ | $\mathbf{538 \pm 70}$ |
| SCOTCH 200, 1k | $400 \pm 17$ | $1370 \pm 29$ | $6425 \pm 80$ | $705 \pm 77$ | $7863 \pm 2670$ |
| $\perp\!\!\!\perp^+_{s,h}$ | $26 \pm 5$ | $80 \pm 8$ | $284 \pm 19$ | $1026 \pm 40$ | $4946 \pm 113$ |
| $\perp\!\!\!\perp_{sym}$+p.p. | $\mathbf{13 \pm 4}$ | $\mathbf{38 \pm 7}$ | $\mathbf{157 \pm 15}$ | $725 \pm 439$ | $4593 \pm 93$ |
| $\perp\!\!\!\perp_{sym}$ only | $57 \pm 84$ | $147 \pm 11$ | $436 \pm 19$ | $1294 \pm 48$ | $6005 \pm 98$ |

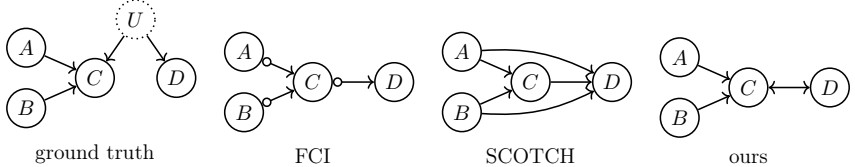

Figure 4: An example graph demonstrating the advantage of our approach under partial observations.

our Algorithm 1 (denoted by $\perp\!\!\!\perp_{s,h}^{+}$) and Algorithm 2 ($\perp\!\!\!\perp_{\mathrm{sym}}$+pp and $\perp\!\!\!\perp_{\mathrm{sym}}$ the result before post-processing) clearly outperform PCMCI and SCOTCH up to $d = 10$. SCOTCH's heavy dependence on hyperparameter choices for $\lambda$ (graph sparsity) and $n_e$ (the number of epochs), can sometimes render it superior when charitably picking the best setting. However, since good values particularly for $\lambda$ cannot be known up front nor selected in a data-driven fashion, one must interpret SCOTCH's performance more conservatively—arguably in terms of its worst case performance over a set of reasonable hyperparameter settings. Hence, SigKer—free of any hyperparameter choices—broadly outperforms the state-of-the-art, such as SCOTCH even in the setting SCOTCH was specifically tailored to (SDE models). Unlike such methods, our approach continues to work well for other data generating mechanisms as our CI test handles stochastic processes beyond the SDE model.

**Summary and discussion of results.** Our CI test for $\perp\!\!\!\perp_{s,h}^{+}$ reliably detects the direction of time and outperforms strong baselines (CCM, Granger, PCMCI), consistently improving with larger sample sizes (Table 1). SigKer also dominates PCMCI in causal discovery across dimensions and consistently beats most hyperparameter settings of SCOTCH. While isolated hyperparameter settings for SCOTCH perform better than SigKer in diffusion dependence and high-dimensional causal discovery, its strong dependence on hyperparameters, particularly the sparsity parameter $\lambda$, renders it unreliable in practice. Finally, SCOTCH and PCMCI are tailored to Markovian SDEs (or SDEs with a fixed lag). Instead, our CI test is broadly applicable and effective across data modalities which we further demonstrate empirically in a functional data example where both PCMCI and SCOTCH fail to detect path-dependence, see Table 10.

**The partially observed setting.** One of the main benefits of constraint-based causal discovery (especially with a non-parametric test) is the ability to handle partial observations. Score-based methods like SCOTCH are fundamentally challenged in this setting, since there could in principle be infinitely many unobserved variables, impossible to model, e.g., with neural-network based approaches. We showcase this advantage on the challenging example graph in Figure 4, where SCOTCH falsely infers an adjacency $A$ — $D$ in 88 out of 100 runs, whereas our algorithm correctly handles the unobserved confounder and only (falsely) predicts this adjacency in 8 instances.

**Real-world pairs trading example.** To demonstrate the applicability of our developed methods on real-data, we evaluate pairs trading strategies on ten stocks from the VBR Small-Cap ETF over a three-year period (2010/01/01–2012/12/31). We only provide a concise summary of our results here and refer to Appendix B.7 for more details on this proof-of-concept study. We assess our method's effectiveness by quantifying the profit-and-loss (P&L)

Table 3: SigKer outperforms baselines in most pairs trading performance metrics. $\uparrow$ / $\downarrow$ indicates 'higher/lower is better'.

|  | return $\uparrow$ | APR $\uparrow$ | Sharpe $\uparrow$ | maxDD $\downarrow$ | maxDDD $\downarrow$ |
|---|---|---|---|---|---|
| ADF | 0.004 | 0.004 | 0.090 | 0.087 | 230 |
| Granger | -0.010 | -0.011 | -0.230 | 0.056 | 219 |
| ADF & Granger | 0.008 | 0.008 | 0.242 | **0.022** | 153 |
| **SigKer** | **0.076** | **0.077** | **1.500** | 0.027 | **21** |

profile of the generated pairs trading strategy as a substitution for ground truth. Pairs were selected based on pairwise p-values from different hypothesis tests: 1) cointegration via the Augmented Dickey-Fuller (ADF) test, 2) Granger causality, and 3) our method. Table 3 shows that our strategy's P&L substantially outperforms the baselines in total return, APR, Sharpe ratio, and maxDDD while being almost on par with ADF & Granger in the fifth relevant metric maxDD. This case-study highlights the broad applicability and potential downstream impact of the findings in this work.

## 5 CONCLUSION AND FUTURE WORK

We introduce constraint based causal discovery algorithms for both fully and partially observed data in the form of stochastic processes, proving them to be sound and complete assuming a CI oracle.

These are based on novel Markov properties corresponding to the developed CI constraints with respect to the induced dependence graph. Our algorithms critically leverage the directionality of time to uniquely identifying the full underlying graph or induced ancestral graph, critically improving over existing constraint-based algorithms in the static case that only output equivalence classes. Our framework also efficiently captures path- or diffusion-dependence. Finally, we propose a practical and consistent kernel-based CI test on path-space leveraging recent advances in signature kernels that empirically outperforms existing alternatives across a wide range of settings also beyond the SDE model such as functional data and fractional Brownian motions.

Due to the identified fundamental limitations in the cyclic setting, we limit ourselves to acyclic (except for self-loops) dependence graphs and assume causal relationships to not change over time. Relaxing both of these assumptions and assessing how much of the causal structure can in principle be learned in the cyclic settings are immediate interesting directions for future work. We highlight that the field of causality represents just one of the many potential applications of our conditional independence test for path-valued random variables. Exploring applications in different domains is a worthwhile direction for future work that could also catalyze new methodological developments.

## REPRODUCIBILITY STATEMENT

Significant effort was made to ensure reproducibility, both for the theoretical and experimental results. The complete proofs of Proposition 3.1, Theorem 3.2, Corollary 3.3 can be found in Appendix A.6 Appendix A.7, Appendix A.10, respectively. Additionally, we discuss the limitations and requirements in Section 1, which outlines the assumptions made throughout the paper. Details regarding implementation and metrics can be found in Section 4, along with further information on the data-generating mechanism and results evaluation throughout Section 4. The implementation of baselines is described in Appendix C, and details on the real-world pairs trading example can be found in Appendix A.11. We will also make all code used to produce the results in this paper openly available.

## ACKNOWLEDGMENTS AND DISCLOSURE OF FUNDING

This work is supported by the DAAD programme Konrad Zuse Schools of Excellence in Artificial Intelligence, sponsored by the Federal Ministry of Education and Research. This work was supported by the Helmholtz Association's Initiative and Networking Fund on the HAICORE@FZJ partition. Emilio Ferrucci is supported by UKRI EPSRC Programme Grant EP/S026347/1. Søren Wengel Mogensen is supported by Independent Research Fund Denmark (DFF-International Postdoctoral Grant 0164-00023B) and a member of the ELLIIT Strategic Research Area at Lund University.

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

APPENDIX

TABLE OF CONTENTS

# A PROOFS AND THEORETICAL DIGRESSIONS

## A.1 SUMMARY OF NOTATIONS

A summary of notation is provided in Table 4.

Table 4: Summary of notations

| Symbol | Meaning |
|---|---|
| $[m]$ | short for $\{1, \ldots, m\}$, $m \in \mathbb{N}$ |
| $X$ | stochastic process $X = (X^1, \ldots, X^d) \in C([0, T], \mathbb{R}^{n_1 + \ldots + n_d})$ |
| $d$ | number of coordinate processes $X = (X^1, \ldots, X^d)$ or nodes in graph |
| $X^k$ | $k$-th coordinate process of $X = (X^1, \ldots, X^d)$, $X_t^k \in \mathbb{R}^{n_k}$ |
| $n_k$ | dimension of the $k$-th coordinate process $X^k$ |
| $W^k$ | $k$-th Brownian motion, $W_t^k \in \mathbb{R}^{m_k}$ |
| $m_k$ | dimension of the $k$-th Brownian motion $W^k$ |
| $n$ | total dimension of the stochastic process $X$, $n = n_1 + \ldots + n_d$ |
| $\mu^k$ | $k$-th drift component $\mu^k : \mathbb{R}^n \to \mathbb{R}^{n_k}$ |
| $\sigma^k$ | $k$-th diffusion component $\sigma^k : \mathbb{R}^n \to \mathbb{R}^{n_k \times m_k}$ |
| $T \in \mathbb{R}_{>0}$ | maximum observation time |
| $t, s \in [0, T]$ | time points within the observation time |
| $\mathcal{G} = (V, E)$ | dependence graph with nodes $V$ and edges $E$ |
| $V \cong [d]$ | set of vertices in the dependence graph |
| $E \subseteq V \times V$ | set of directed edges between nodes $V$ |
| $\mathrm{pa}_v^{\mathcal{G}}$ | parents of node $v$ in graph $\mathcal{G}$ (omitted if clear from context) |
| $\tilde{\mathcal{G}} = (\tilde{V}, \tilde{E})$ | lifted dependence graph of graph $\mathcal{G} = (V, E)$ |
| $V_0, V_1$ | disjoint copies of $V$, with $0/1$ indicating past/future |
| $\tilde{V} := V_0 \sqcup V_1$ | node set in lifted dependence graph |
| $\tilde{E} \subseteq \tilde{V} \times \tilde{V}$ | edge set in lifted dependence graph |
| $S(X)_{s,t} \in \mathcal{T}(\mathbb{R}^d)$ | signature transform of path $X$ over interval $[s, t]$ |
| $K_S(X, Y)$ | signature kernel of paths $X, Y$ |
| $T(\mathbb{R}^d)$ | the direct sum $\bigoplus_{n=1}^{\infty} (\mathbb{R}^d)^{\otimes n} := \{a = (a_n)_{n \in \mathbb{N}} = \prod_{n \in \mathbb{N}} (\mathbb{R}^d)^{\otimes n}; \max\{n : a_n \neq 0\} < \infty\}$ |
| $\mathcal{T}(\mathbb{R}^d)$ | the completion of $T(\mathbb{R}^d)$ w.r.t. $\langle a, b \rangle = \sum_{n=1}^{\infty} a_n b_n$ (well-defined) |
| $\mathcal{O}(\cdot)$ | Landau asymptotic notation ("Big-O notation") |

## A.2 INTUITION AND DETAILS FOR THE SDE MODEL

Making the actual dependence of $\mu^k$ and $\sigma^k$ on their arguments in eq. (1) explicit, we can rewrite it as

$$\begin{cases} \mathrm{d}X_t^k = \mu^k(X_{[0,t]}^{\mathrm{pa}_k})\mathrm{d}t + \sigma^k(X_{[0,t]}^{\mathrm{pa}_k})\mathrm{d}W_t^k, \\ X_0^k = x_0^k \qquad k \in [d]. \end{cases} \tag{7}$$

Then $\mu^k, \sigma^k$ are functions defined on $C([0, +\infty), \mathbb{R}^{\dim(\mathrm{pa}_k)})$ (or some suitable subspace thereof): Lipschitz conditions on the coefficients that guarantee existence and uniqueness for this type of SDE can be found in Rogers & Williams (2000), which we assume to hold throughout. Each $W^k$ is an $m_k$-dimensional Brownian motion (a collection of $m_k$ independent 1-dimensional Brownian motions, that is), $\sigma^k$ maps to the space of $n_k \times m_k$-dimensional matrices, and the noises $W^k$ together with the (possibly random) initial conditions $x_0^k$ are jointly independent. In other words, the system can be written as an $n := n_1 + \ldots + n_d$-dimensional SDE driven by an $m := m_1 + \ldots + m_d$-dimensional Brownian motion, with a block-diagonal diffusion coefficient $\sigma$ (since the noise is unobserved, this structure has to be imposed if we wish not to deal with unobserved confounding). This means the SDE can be written in the matrix-form

$$\mathrm{d}\begin{pmatrix} X_t^1 \\ \ldots \\ X_t^d \end{pmatrix} = \begin{pmatrix} \mu^1(X_{[0,t]}^{\mathrm{pa}_1}) \\ \ldots \\ \mu^d(X_{[0,t]}^{\mathrm{pa}_d}) \end{pmatrix} \mathrm{d}t + \begin{pmatrix} \sigma^1(X_{[0,t]}^{\mathrm{pa}_1}) & \cdots & 0 \\ \vdots & \ddots & \vdots \\ 0 & \cdots & \sigma^d(X_{[0,t]}^{\mathrm{pa}_d}) \end{pmatrix} \mathrm{d}\begin{pmatrix} W_t^1 \\ \ldots \\ W_t^d \end{pmatrix} \tag{8}$$

The superscript $\mathrm{pa}_k$ refers to the parents of the $k^{\text{th}}$ node in $\mathcal{G}$, which may (and most often does) contain $k$ itself. Intuitively, all of this means that, for all times $t$ and small increments $\Delta t$, the increment of the solution $X_{t+\Delta t}^k - X_t^k$ is random with distribution that is well-approximated by a multivariate normal with mean $\mu^k(X_{[0,t]}^{\mathrm{pa}_k})\Delta t$ and covariance function $\sigma^k(X_{[0,t]}^{\mathrm{pa}_k})\sigma^k(X_{[0,t]}^{\mathrm{pa}_k})^\intercal \Delta t$, and independent of the history of the system up to time $t$. This interpretation can actually be made precise by showing that these piecewise constant paths converge in law to the true solution (usually known as the weak solution, when viewed in this way).

We will refer to $\mathcal{G}$ as the *dependence graph* of the eq. (1), which we refer to as the *SDE model* for brevity. Compared to that of Peters et al. (2022), this model is slightly more general in that (i) it allows for path-dependence and (ii) for each node to represent a multidimensional process; the special case of state-dependent SDE—i.e., in which $\mu$ and $\sigma$ only depend on $X_t$, the value of $X$ at time $t$—continues to be an important special case, although our model also accommodates delayed SDEs (the coefficients depend on the value of $X$ at a prior instant in time, e.g., $X_{t-\tau}$ for fixed or possibly random/time-dependent $\tau > 0$), and SDEs that depend on quantities involving the whole past of $X$, such as the average $t^{-1}\int_0^t X_s \mathrm{d}s$. We note that the choice not to make the coefficients explicitly time-dependent is deliberate and motivated by the requirement that the system be *causally stationary* (the causal relations between the variables do not change over time).

Given that we are considering the dynamic setting, it is generally natural to allow for $\mathcal{G}$ to have cycles. This comes at no additional requirement of consistency constraints as it does in the static case (see for example Bongers et al. (2021)), since the causal arrows in the model eq. (1) should be thought of as 'pointing towards the infinitesimal future, with infinitesimal magnitude', integrated over the whole time interval considered. Indeed, the system of SDEs does not require any global consistency to be well-posed, other than the regularity and growth conditions that guarantee global-in-time existence and uniqueness. On the other hand, we will be interested in the potential for constraint-based causal discovery of such systems, qualified by the following assumption on the types of conditional independencies that we allow to be tested in continuous time:

### A.3 COUNTEREXAMPLE FOR CYCLIC CAUSAL DISCOVERY IN THE SDE MODEL

The following example demonstrates that constraint-based causal discovery of the full graph is impossible using expressions of the form eq. (2) when allowing for cycles.

**Example A.1.** An oracle for eq. (2) is not powerful enough to rule out a directed edge $X^1 \to X^3$ for an SDE model with the following dependence graph:

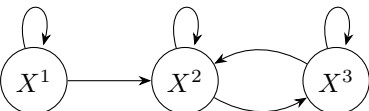

We now go through this example and describe a different type of oracle that would be required for causal discovery in cyclic SDE models. Concretely, we claim that an oracle for eq. (2) is not powerful enough to rule out a directed edge $X^1 \to X^3$ for an SDE model with the graph in Example A.1. Testing $X_{[0,s]}^1 \perp\!\!\!\perp X_{[s,s+h]}^3 \mid X_{[0,s]}^{2,3}$, will not remove the edge, due to the open path

$$X_{[0,s]}^1 \to X_{[s,s+h]}^1 \to X_{[s,s+h]}^2 \to X_{[s,s+h]}^3.$$

Testing $X_{[0,s]}^1 \perp\!\!\!\perp X_{[s,s+h]}^3 \mid X_{[0,s]}^3, X_{[0,s+h]}^2$, on the other hand, runs into the collider

$$X_{[0,s]}^1 \to X_{[s,s+h]}^1 \to X_{[s,s+h]}^2 \leftarrow X_{[s,s+h]}^3.$$

There is no other way of splitting up the time interval (even allowing for more than 2 subintervals) that overcomes both these problems: when testing $X_{[0,s]}^1 \perp\!\!\!\perp X_{[s,s+h]}^3$ (as is necessary in order to rely on time to obtain direction of the arrow), if $X^2$ is not conditioned on over $[s, s + h]$ it will be a mediator, and if it is, it will be a collider.

What would be needed to detect the edge in the above example (and to perform causal discovery more generally in the cyclic case), is the availability of tests of *strong instantaneous non-causality*

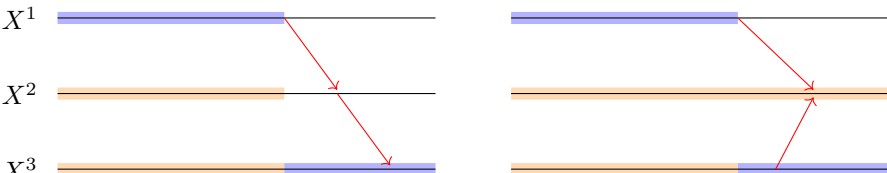

Figure 5: The blue highlighter is for intervals over which the path is being tested for independence, while orange is for conditioning. The first figure illustrates the path opened by the unconditioned-on mediator $X^2_{[s,s+h]}$, while the second illustrates the path opened by the same variable acting as a conditioned-on collider. While not quite possible to draw extra/figures this way, it is helpful to think of the arrows as pointing from values to increments $X_t \rightarrow (Y_{t+dt} - Y_t)$, infinitesimally, with the total causal effect accrued over time.

*in the Granger sense* as defined in Florens & Fougere (1996). This reflects the fact that the SDE model is 'acyclic at infinitesimal scales'. Unfortunately, such a property—which has to do with the Doob-Meyer decompositions of functions of the process w.r.t. two different filtrations—is much more difficult to test for, as it is infinitesimal in nature. Recent progress in this direction was made in Christgau et al. (2023) for the case of SDEs driven by jump processes; this local independence criterion, which goes back to Schweder (1970), is however *weak* in the sense that it only takes into account the Doob-Meyer decomposition of $X$ and not functions of it, and is not therefore able, for example, to detect dependence in the diffusion coefficient (Gégout-Petit & Commenges, 2010). Here we take the view that a continuously-indexed path, queried over an interval, can be considered as an observational distribution (cf. the discussion in Sokol & Hansen (2013) on whether the infinitesimal generator can be considered 'observational').

### A.4    Signature Kernel Details

**Additional mathematical details.**    For any $p \in [1, 2)$, let $\mathcal{K} \subset C_p([0, T], \mathbb{R}^d)$ be a compact subset such that the signature transform is a continuous injection from $\mathcal{K}$ to $\mathcal{T}(\mathbb{R}^d)$. An example of compact set $\mathcal{K}$ satisfying such conditions is the set of time-augmented paths started at a common origin, where $C_p([0, T], \mathbb{R}^d)$ is endowed with the $p$-variation topology. Further examples of compact sets and choices of topologies on path-space are discussed by Cass & Turner (2024). Then, the signature kernel is *universal* over $\mathcal{K}$ in the sense that the associated RKHS, whose elements are restricted to act on $\mathcal{K}$, is dense in $C(\mathcal{K})$, the space of continuous real-valued functions on $\mathcal{K}$, endowed in the topology of uniform convergence. See Cass et al. (2023, Proposition 3.3) for a proof. The compactness assumption can be relaxed by either renormalizing the signature transform so that its range is always guaranteed to be bounded (Chevyrev & Oberhauser, 2022) or by operating on weighted path-spaces with a different choice of topology Cuchiero et al. (2023).

The signature kernel is well-defined and can be shown, under similar compactness conditions, to be universal also when $p \geq 2$ (Salvi et al., 2021a; Lemercier & Lyons, 2024). In the specific setting of stochastic differential equations (SDEs) considered in this paper, i.e. when $p \in [2, 3)$, one can use a classical limiting procedure from rough analysis to show that the integral equation in eq. (4) takes the Stratonovich form $f(t, t') = 1 + \int_s^t \int_{s'}^{t'} f(u, v) \langle \circ dX_u, \circ dY_v \rangle_1$. We refer the interested reader to Salvi et al. (2021a, Section 4.3) for further details.

Another important property that a kernel should possess in order to ensure theoretical guarantees of most kernel methods, including tests for conditional independence (Muandet et al., 2017), is that of being *characteristic*. Loosely speaking, a kernel $k$ is said characteristic to a topological space if for any (Borel) probability measure $\mu$ on that space, the $\mu$-expectation of the feature map $k(x, \cdot)$ uniquely characterises $\mu$. An important result from the theory of kernels relates to the equivalence between the notions of universality and characteristicness. In particular, these three notions can be shown to be equivalent for kernels defined over general locally convex topological vector spaces (Simon-Gabriel & Schölkopf, 2018, Theorem 6).

**Intuition.**    The signature transform of a (potentially multidimensional) continuous path (or time series) is a sequence of iterated integrals, static features, which form an expressive representation

of the path. An analogy is often drawn to monomials in (multiple) indeterminants, with which one can express continuous functions. Another useful analogy is the Fourier transform. While Fourier transforms (or wavelet transforms) capture the frequency content of different modes (over different times), the signature captures information about order, area, and so forth (due to the *iterated integrals*). For example, the depth 1 (or order 1) iterated integral describes the changes of each variable throughout the time interval; depth 2 describes the signed area that is swept by the curve (an illustration of depth 1 and 2 of a path can be found in Morrill et al. (2020)); higher depths capture properties of the volume, and so on, but those can be hard to grasp visually. We refer the reader to Chevyrev & Kormilitzin (2016); Lee & Oberhauser (2023) as introductory texts on the signature (kernel) method that contain more visualizations and examples for building intuitions about these mathematical objects.

## A.5 Testing on Time Points

It is also possible to replace any of the path-valued random variables in eq. (2) by a random variable at a single time-point (as for example done in the post-processing by testing on $X_0$) using a regular rbf-kernel. Note, that in addition, if we knew in advance that there is no path-dependence, the unconditional independence $X^i \perp\!\!\!\perp^+_{s,h} X^j$ is equivalent to the conditional independence $X^i_s \perp\!\!\!\perp X^j_{s+h} - X^j_s \mid X^j_s$ (at fixed times $s, s+h$): this is because without path-dependence $X^j_u$ for $u \in [s,t]$ only depends on $X^j_{[s,t]}$ via $X^j_s$. Testing on entire (segments of) paths is always required when allowing for path-dependence.

## A.6 Proof of Proposition 3.1

*Proof of Proposition 3.1.* Since $\mathcal{G}$ is acyclic, a directed cycle in $\tilde{\mathcal{G}}$ that is not a loop must contain nodes both in $V_0, V_1$. But there can be no such directed cycles, since edges only travel in the direction $V_0 \to V_1$, by construction of $\tilde{\mathcal{G}}$. Since $\tilde{\mathcal{G}}$ also free of loops, it is a DAG. We now show that there exists a structural causal model (SCM) over $\tilde{\mathcal{G}}$ with the path-valued random variables of the statement: Markovianity will then follow from Peters et al. (2017, Proposition 6.31), original to Verma & Pearl (1990). In other words, we must show there exist Borel-measurable functions

$$F_0^k : \begin{cases} \mathbb{R}^{\dim(\mathrm{pa}_k^{\mathcal{G}})} \times C([0,s], \mathbb{R}^{m_k}) \times C([0,s], \mathbb{R}^{\dim(\mathrm{pa}_k^{\mathcal{G}} \setminus \{k\})}) & \rightharpoonup C([0,s], \mathbb{R}^{n_k}) \\ (x_0^{\mathrm{pa}_k^{\mathcal{G}}}, W_{[0,s]}^k; X_{[0,s]}^{\mathrm{pa}_k^{\mathcal{G}} \setminus \{k\}}) & \mapsto X_{[0,s]}^k \end{cases}$$

and

$$F_1^k : \begin{cases} C([s,s+h], \mathbb{R}^{m_k}) \times C([0,s], \mathbb{R}^{\dim(\mathrm{pa}_k^{\mathcal{G}})}) \times C([s,s+h], \mathbb{R}^{\dim(\mathrm{pa}_k^{\mathcal{G}} \setminus \{k\})}) & \rightharpoonup C([s,s+h], \mathbb{R}^{n_k}) \\ ((W_{s+t}^k - W_s^k)_{0 \le t \le h}; X_{[0,s]}^{\mathrm{pa}_k^{\mathcal{G}}}, X_{[s,s+h]}^{\mathrm{pa}_k^{\mathcal{G}} \setminus \{k\}}) & \mapsto \tilde{X}_{[s,s+h]}^k \end{cases}$$

for each $k \in [d]$ where the initial conditions $x_0$ and the Brownian path segments $W_{[0,s]}^k, (W_{s+t}^k - W_s^k)_{0 \le t \le h}$ are jointly independent and $\tilde{X}_{[s,s+h]}^k := X_{[s,s+h]}^k - X_s^k$. This independence follows from the definition of the SDE model and from independence of Brownian increments over disjoint or consecutive intervals. These functions are partial in that they are not defined on the whole space of continuous functions on the interval: we only need to show them to be defined on a measurable set of paths that contains all solutions of SDEs on the required interval. $F_1^k$ can be defined as follows, with care to make explicit the dependence on the solution on the two intervals via the operation of path-concatenation $*$:

$$X_{[s,s+h]}^k = \text{solution of } X_u^k = X_s^k + \int_s^u \mu^k(X_{[0,s]}^{\mathrm{pa}_k^{\mathcal{G}} \setminus \{k\}} * X_{[s,u]}^{\mathrm{pa}_k^{\mathcal{G}} \setminus \{k\}}, X_{[0,s]}^{\{k\} \cap \mathrm{pa}_k^{\mathcal{G}}} * X_{[s,u]}^{\{k\} \cap \mathrm{pa}_k^{\mathcal{G}}}) \mathrm{d}t \quad (9)$$

$$+ \int_s^u \sigma^k(X_{[0,s]}^{\mathrm{pa}_k^{\mathcal{G}} \setminus \{k\}} * X_{[s,u]}^{\mathrm{pa}_k^{\mathcal{G}} \setminus \{k\}}, X_{[0,s]}^{\{k\} \cap \mathrm{pa}_k^{\mathcal{G}}} * X_{[s,u]}^{\{k\} \cap \mathrm{pa}_k^{\mathcal{G}}}) \mathrm{d}W_t^k$$

$$(10)$$

for $u \in [s, s+h]$, we obtain $F_1^k$ by removing the initial condition $X_s^k$ in front of the integrals in equation 9.

We have also separated out the $k$-th component of the solution from the rest of the arguments on which $\sigma^k, \mu^k$ are dependent, which we consider part of the measurable adapted dependence on $W^{\mathrm{pa}\setminus\{k\}}$ needed in the existence and uniqueness theorem. $F_0^k$ is defined similarly (with no dependence on past path-segments and with $x_0^k$ replacing $X_s^k$ in equation 9). Such functions are well-known to be well-defined and measurable (Rogers & Williams, 2000, Theorem 10.4). $\qquad\square$

*Remark* A.2. In testing, using the symmetric criterion $\perp\!\!\!\perp_{\mathrm{sym}}$ has proven to be more reliable than the asymmetric one used in the test above. A procedure, which is a little redundant, but which makes good use of the superior performance of $\perp\!\!\!\perp_{\mathrm{sym}}$ would work as follows:

1. Run the PC algorithm to discover the CPDAG corresponding to the variables $X_{[0,T]}^k$.

2. Pass this CPDAG as input to Algorithm 1: by this we mean that $\tilde{V}$ is the same and the input $\tilde{E}$ is given by

$$\{i_0 \to j_0, \ i_1 \to j_1, i_0 \to j_1 \mid i, j \in V, \ (i \to j) \text{ or } (i - j) \in E'\} \cup \{k_0 \to k_1 \mid k \in V\}$$

where $(i \to j)$ denotes an undirected arrow and $E'$ is the CPDAG output by the classical PC algorithm.

3. The loop-recovery step is identical as before.

The algorithm would run very similarly as before, but performing fewer CI tests in the first phase, since it can make use of the information contained in the partially-directed skeleton.

*Remark* A.3. If we knew in advance that there is no path-dependence, the test $X^i \perp\!\!\!\perp_{s,h}^+ X^j$ (with no additional conditioning set) is equivalent to the test on values of the process $X_s^i \perp\!\!\!\perp X_{s+h}^j \mid X_s^j$: this is because $X_u^j$, for $u \in [s, t]$ only depends on $X_{[s,t]}^j$ via $X_s^j$. However, this kind of statement no longer works in the conditional case (by arguments similar to Example A.1), and testing on whole paths is strictly necessary when allowing for path-dependence.

*Remark* A.4. CI testing of SDEs is conceptually similar to static CI on discretely sampled data, when the sampling rate is lower than the actual frequency at which causal effects propagate. This is because, in continuous time and for any $h > 0$, there are going to be causal effects that occur at time scales less than $h$. Whenever we just discretely sample time series with no instantaneous effects, Peters et al. (2017, Thm. 10.3) provides full causal discovery, even in the presence of cycles in the summary graph, by performing $d^2$ tests $X_s^i \perp\!\!\!\perp X_{s+\Delta s}^j \mid X_{[0,s]}^{[d]\setminus\{i,j\}}$. Potentially absent loops could then be removed as in Algorithm 1. Ignoring that this might suffer from the conditioning set being too large, our signature method can be of help in this setting too, if there is path-dependence: the conditioning variable can be taken to be $S(X^{[d]\setminus\{i,j\}})_{0,s}$.

## A.7 PROOF OF THEOREM 3.2

*Proof of Theorem 3.2.* We have to show that $E^{Alg} = E$ (where $E^{Alg}$ the edges detected by Algorithm 1).

"$\subseteq$:" This direction can be shown by finding an appropriate separation set in $\tilde{\mathcal{G}}$ by exploiting the global Markov property Proposition 3.1. We begin by proving correctness of the recovery of the skeleton modulo loops. This will follow if we show that for $(i \to j) \notin E$ for $i, j \in V$ distinct

$$i_0 \perp\!\!\!\perp_{d-sep}^{\tilde{\mathcal{G}}} j_1 \mid \{j_0\} \cup \{k_0, k_1 \mid k \in \mathrm{pa}_j^{\mathcal{G}} \setminus \{j\}\},$$

where $\perp\!\!\!\perp_{d-sep}^{\tilde{\mathcal{G}}}$ denotes $d$-separation in the graph $\tilde{\mathcal{G}}$. Indeed, eventually, the set $K$ in the algorithm will take the value $\{k_0, k_1 \mid k \in \mathrm{pa}_j^{\mathcal{G}} \setminus \{j\}\}$, at which point the three edges $(i_0, j_0), (i_1, j_1), (i_0, j_1)$ (which are either all present or all absent, inductively, by construction of $\tilde{E}$) are deleted. (Of course it may be that these edges are deleted before this, if a smaller or different $d$-separating set is found, but note that edges are never added.) All undirected paths between $i_0$ and $j_1$ factor as $i_0 \cdots h_0 \to l_1 \cdots j_1$, where the dots stand for a possibly empty undirected path. All such paths with $l_1 = j$ are blocked by $\{j_0\} \cup \{k_0 \mid k \in \mathrm{pa}_j^{\mathcal{G}} \setminus \{j\}\}$, so we focus our attention on the case in which $l_1 \cdots j_1$ is non-empty.

Here we follow a similar argument to that of Verma & Pearl (1990, Lemma 1). If $l_1 \cdots j_1$ is of the form $l_1^1 \cdots l_1^r \to j_1$ (with $l_1^1 = l_1$) it is blocked by $\{k_1 \mid k \in \mathrm{pa}_j^{\mathcal{G}} \setminus \{j\}\}$. Assume instead it is of the form $l_1^1 \cdots l_1^r \leftarrow j_1$ and let $1 \leq q \leq r$ be such that $l_1^q$ is the first collider on the entire path $i_0 \cdots j_1$, starting from $j_1$ and travelling back: certainly this exists (and belongs to $V_1$), thanks to the arrows $h_0 \to l_1^1$ and $l_1^r \leftarrow j_1$. For the path $i_0 \cdots j_1$ to be open given the conditioning set, it must be the case that $l_1^p$ must be an ancestor (in $\tilde{\mathcal{G}}$) or member of $\{k_1 \mid k \in \mathrm{pa}_j^{\mathcal{G}} \setminus \{j\}\}$ ($l_1^p$ cannot be an ancestor of nodes in $V_0$): this would yield a directed cycle, which is impossible.

We argue similarly for the loop-removal phase. If $(i \in i) \notin E \iff (i_0, i_1) \notin \tilde{E}$, we must show

$$i_0 \perp\!\!\!\perp_{d-sep}^{\tilde{\mathcal{G}}} i_1 \mid \{k_0, k_1 \mid k \in \mathrm{pa}_i^{\mathcal{G}} \setminus \{i\}\}$$

Consider a path $i_0 \cdots h_0 \to l_1 \cdots i_1$ in $\tilde{\mathcal{G}}$. If the segment $l_1 \cdots i_1$ is non-empty we conclude that the path is blocked by the same argument as above (with $i_1$ replacing $j_1$). Assume that the path is of the form $i_0 \cdots h_0 \to i_1$ (with $i_0 \neq h_0$ since $(i_0, i_1) \notin \tilde{E}$): in this case $h \in \mathrm{pa}_i^{\mathcal{G}} \setminus \{i\}$ and the path is again blocked. Thus all paths are blocked and the implication follows.

$\supseteq$: Assume $(i, j) \in E$, thus $(i_0, j_1) \in \tilde{E}$. Hence as $\tilde{\mathcal{G}}$ a DAG by construction, $\nexists S \subseteq V \setminus \{i, j\}$ s.t. $\{i_0\} \perp\!\!\!\perp_d^{\tilde{\mathcal{G}}} \{j_1\} \mid S_0 \cup \{j_0\} \cup S_1$. By the faithfulness assumption Definition A.5 one therefore has $X_{[0,s]}^i \not\perp\!\!\!\perp X_{[s,s+h]}^j \mid X_{[0,s]}^S, X_{[0,s]}^j, X_{[s,s+h]}^S \; \forall S \subseteq V \setminus \{i, j\}$. Hence $(i, j) \in E^{Alg}$. $\qquad\square$

## A.8 Faithfulness and Causal Minimality

In the above proof, we have used the following notion of *faithfulness*:

**Definition A.5** ((Global) Faithfulness of $\perp\!\!\!\perp_{s,h}^+$ in $\tilde{\mathcal{G}}$). Let $\{X_t^i\}_{i \in [d]}$ be the coordinate processes of a solution of eq. (1) for $t \in [0, 1]$, $\mathcal{G} = (V \cong [d], E)$ the dependence graph defined by eq. (1). The independence relation $\perp\!\!\!\perp_{s,h}^+$ is called *(globally) faithful* w.r.t. the DAG $\tilde{\mathcal{G}}$, if

$$X^A \perp\!\!\!\perp_{s,h}^+ X^B \mid X^C \;\Rightarrow\; A_0 \perp\!\!\!\perp_d^{\tilde{\mathcal{G}}} B_1 \mid B_0, C_0, C_1 \tag{11}$$

for all $A, B, C \subseteq V$ pairwise disjoint.

Faithfulness is a standard assumption in the causal discovery literature. Some type of faithfulness-like assumption is strictly necessary to do causal discovery since, otherwise, it would be impossible to make any inference about adjacencies in the underlying graph from the observed distribution. While we used the strong faithfulness assumption Definition A.5 for the proof above, various weaker faithfulness-like assumptions in the case of DAG-based causal discovery could be used, e.g., the strictly weaker assumption known as 'parent faithfulness' (Mogensen (2024) gives a definition of parent faithfulness in the context of stochastic processes). This assumption suffices in our algorithm without making any changes to it (or the respective proofs) in the acyclic case.

The strength of the faithfulness assumption in real-world data is heavily debated. Besides Weinberger (2018), who states that 'proposed counterexamples to the causal faithfulness condition are not genuine', Spirtes et al. (2000) argues that for parametric models, 'the measure of the set of free parameter values for any DAG that lead to such cancellations is zero for any smooth prior probability density, such as the Gaussian or exponential one, over the free parameters'. What is meant in our case is, that when allowing the diffusion coefficients $\sigma$ in eq. (1) to range in some 'reasonable' finite-dimensional family (such as linear or affine functions) and drawing the parameters of such a family from a distribution which has positive density w.r.t. the Lebesgue measure (and independently of the noise), the resulting distribution on path-space, split over $[0, s]$ and $[s, t]$, will almost surely be faithful for any $0 < s < t$ w.r.t. the lifted dependence graph. We do not attempt a proof of this statement here, as it may require considerable effort and technique. On the other side, Andersen (2013) or Uhler et al. (2013) (among others) caution against making the strong faithfulness assumption easily. Many of their concerns are ameliorated by only assuming the weaker parent faithfulness.

Causal minimality, on the other hand, is generally easier to understand and can be expected to hold for the arrow $X_{[0,s]}^i \to X_{[s,t]}^j$ whenever the following condition is satisfied: the conditional distribution $X_u^j \mid X^{\mathrm{pa}_j^{\mathcal{G}} \setminus \{i\}} = x^j$ admits a positive density on some submanifold $M_u^x$ of $\mathbb{R}^n$ and for all $u$ there exist $x_1^i, x_2^i \in M_u^x$, $x_1^i \neq x_2^i$, s.t. $(\mu^j, \sigma^j)(x, x_1^i) \neq (\mu^j, \sigma^j)(x, x_2^i)$.

**Example A.6** (Causal minimality). The fact that we are allowing for variables to have more than one dimension means the causal minimality condition is not a trivial one, as the following examples demonstrate. Let $X^1 = (W, 0)$ where $W$ is a 1-dimensional Brownian motion (so that $n_1 = 2$), and let $\mu^1$ and/or $\sigma^1$ depend on $X^1$ only through its second coordinate. Then, even though this dependence may be non-trivial, causal minimality does not hold. This may lead one to believe that one can generically only expect causal minimality to hold if $m = n$, but this is not the case. Take, for example, $d = 4 = n$ and assume $\sigma^3 \equiv 0$, i.e., $X^3$ has no driving Brownian motion, so that $m$ is only 3. Assume, furthermore, that $\text{pa}_4 = \{4\}$. Then, by Hörmander's Lie bracket-generating condition (see for example §V.38 in Rogers & Williams (2000)), $(X^1, X^2, X^3)$ has a density in $\mathbb{R}^3$ for any choice of an initial condition, if $(\sigma^1(x), 0, 0), (0, \sigma^2(x), 0)$ span $\mathbb{R}^2$ for all $x \in \mathbb{R}^3$ and $[(\sigma^1(x), 0, 0), \mu] = \sigma^1 \partial_1 \mu - \mu^i \partial_i \sigma^1$ or $[(\sigma^2(x), 0, 0), \mu]$ span the third direction. Arrows going from the first three nodes to the fourth will be necessary for Markovianity as long as the coefficients of $X^4$ depend on the respective variables on the support of such density. The point is that even though $(X^1, X^2, X^3)$ is only driven by a two-dimensional Brownian motion, the coefficients of the SDE (at the initial condition) impart a 'twist' to the solution, making causal minimality a trivial condition on $(\sigma^4, \mu^4)$ which is verified as soon as there is any dependence.

## A.9 GLOBAL MARKOV PROPERTY FOR THE SYMMETRIC INDEPENDENCE CRITERION

In this section, we provide a proof for the global Markov property of the symmetric independence criterion in order for the PC-algorithm to be applicable. It is based on the notion of independence models:

**Definition A.7.** Let $V \cong [d]$ be a set. An *independence model* $\mathcal{J}(V)$ over $V$ is a ternary relation over disjoint subsets of $V$,

$$\mathcal{J}(V) \subset \{(A, B, C) \mid A, B, C \subset V \text{ disjoint}\}$$

When $(A, B, C)$ is a triple in $\mathcal{J}(V)$, $(A, B, C) \in \mathcal{J}(V)$, we also use $\langle A, B \mid C \rangle$ to denote $(A, B, C)$. This notation highlights the fact that $C$ is a conditioning set. An independence model $\mathcal{J}(V)$ is called a *semigraphoid* if 1.-4. hold for all disjoint $A, B, C \subset V$.

1. (symmetry) $\langle A, B \mid C \rangle \in \mathcal{J}(V) \Rightarrow \langle B, A \mid C \rangle \in \mathcal{J}(V)$

2. (decomposition) $\langle A, B \mid C \rangle \in \mathcal{J}(V)$ and $D \subset B \Rightarrow \langle A, D \mid C \rangle \in \mathcal{J}(V)$

3. (weak union) $\langle A, B \cup D \mid C \rangle \in \mathcal{J}(V) \implies \langle A, B \mid C \cup D \rangle \in \mathcal{J}(V)$

4. (contraction) $\langle A, B \mid C \rangle \in \mathcal{J}(V)$ and $\langle A, D \mid B \cup C \rangle \in \mathcal{J}(V) \implies \langle A, B \cup D \mid C \rangle \in \mathcal{J}(V)$

A semigraphoid $\mathcal{J}(V)$ is called *graphoid* if 5. holds.

5. (intersection) $\langle A, B \mid C \cup D \rangle \in \mathcal{J}(V)$ and $\langle A, D \mid B \cup C \rangle \in \mathcal{J}(V) \implies \langle A, B \cup D \mid C \rangle \in \mathcal{J}(V)$

An independence model $\mathcal{J}(V)$ called *compositional* if 6. holds.

6. (composition) $\langle A, B \mid C \rangle \in \mathcal{J}(V)$ and $\langle A, D \mid C \rangle \in \mathcal{J}(V) \implies \langle A, B \cup D \mid C \rangle \in \mathcal{J}(V)$

The graphical criterion of d-separation defines a independence model on a DAG $\mathcal{G} = (V, \mathcal{E})$ by

$$\mathcal{J}_{d-sep}(\mathcal{G}) = \{(A, B, C) \mid A, B, C \subset V \text{ disjoint}, A \perp\!\!\!\perp_{d-sep} B \mid C\} \tag{12}$$

**Theorem A.8** (Lauritzen & Sadeghi (2018)). *For any DAG $\mathcal{G} = (V, \mathcal{E})$, the independence model $\mathcal{J}_{d-sep}(\mathcal{G})$ is a compositional graphoid.*

Let $V = [d]$. Given random variables $X^i : (\Omega, \mathcal{A}, P) \to (\mathcal{X}^i, \mathcal{A}_i)$, $i \in V$, and $A, B, C \subset V$ disjoint, conditional independence

$$X^A \perp\!\!\!\perp X^B \mid X^C :\Leftrightarrow \sigma(X^A) \perp\!\!\!\perp \sigma(X^B) \mid \sigma(X^C)$$

defines the *probabilistic independence model*

$$\mathcal{J}(P) = \{(A, B, C) \mid A, B, C \subset V \text{ disjoint}, X_A \perp\!\!\!\perp X_B \mid X_C\}. \tag{13}$$

We will use this as a *symmetric conditional independence* relation. Note that $X^i$ may take values in a path-space in which case it is a stochastic process. The independence model $\mathcal{J}(P)$ is a semigraphoid which is an immediate consequence of sub-$\sigma$-algebra properties.

For $v \in V$, we let $\mathrm{nd}_v^{\mathcal{G}}$ denote the set of *nondescendants* of $v$, i.e., the set of nodes $i$ such that there is no directed path from $v$ to $i$.

**Definition A.9** (Directed global and local Markov properties). Let $\mathcal{G} = (V, \mathcal{E})$ be a DAG and $\mathcal{J}(V)$ be an independence model over $V$. The independence model $\mathcal{J}(V)$ satisfies the *global Markov property w.r.t. $\mathcal{G}$* :$\Leftrightarrow$

$$\langle A, B \mid C \rangle \in \mathcal{J}_{d-sep}(\mathcal{G}) \quad \implies \quad \langle A, B \mid C \rangle \in \mathcal{J}(V) \qquad \forall A, B, C \subset V \text{ disjoint} \tag{14}$$

The independence model $\mathcal{J}(V)$ satisfies the *directed local Markov property w.r.t. $\mathcal{G}$* :$\Leftrightarrow$

$$\langle \{v\}, \left( \mathrm{nd}_v^{\mathcal{G}} \backslash \mathrm{pa}_v^{\mathcal{G}} \right) \mid \mathrm{pa}_v^{\mathcal{G}} \rangle \in \mathcal{J}(V) \text{ for all } v \in V \tag{15}$$

**Theorem A.10** (Lauritzen et al. (1990)). *Let $\mathcal{G} = (V, \mathcal{E})$ be a DAG, let and $\mathcal{J}(V)$ a semigraphoid over $V$. The directed global and local Markov properties are equivalent, that is,*

$$\mathcal{J}(V) \quad \text{satisfies eq. (14) w.r.t. } \mathcal{G} \Leftrightarrow \mathcal{J}(V) \quad \text{satisfies eq. (15) w.r.t. } \mathcal{G} \tag{16}$$

We therefore only have to establish the local Markov property for the symmetric independence model $\perp\!\!\!\perp_{\mathrm{sym}}$:

**Proposition A.11.** *Let $X$ be a set of variables induces by the model in eq. (7) with a constant and diagonal diffusion matrix such that it induces the DAG $\mathcal{G} = (V, \mathcal{E})$. Then the system satisfies the* directed local Markov property *w.r.t. $\mathcal{G}$,*

$$X^i \perp\!\!\!\perp_{\mathrm{sym}} \mathrm{nd}_{X^i}^{\mathcal{G}} \setminus \mathrm{pa}_{X^i}^{\mathcal{G}} \mid \mathrm{pa}_{X^i}^{\mathcal{G}} \qquad \forall i \in [d] \tag{17}$$

*Proof.* The idea of the proof is that all information about node $X^i$ is contained in its parents $pa_i := \mathrm{pa}_{X^i}^{\mathcal{G}}$, the Brownian motion $W^i$, and the initial condition $X_0^i$. A similar idea can be used to show the result in the case of a SCM Pearl (2009). We let $\tilde{\mathcal{J}}(P)$ denote the semigraphoid induced by $\perp\!\!\!\perp_{\mathrm{sym}}$ on the variable set $\tilde{V} = \{X^1, \dots, X^n\} \cup \{W^1, \dots, W^n\} \cup \{X_0^1, \dots, X_0^n\}$, and we let $\mathcal{J}(P)$ denote the semigraphoid induced by $\perp\!\!\!\perp_{\mathrm{sym}}$ on the variable set $V = \{X^1, \dots, X^n\}$. We define the DAG

$$\tilde{\mathcal{G}} = \left( \tilde{V} = \{X^1, \dots, X^n\} \cup \{W^1, \dots, W^n\} \cup \{X_0^1, \dots, X_0^n\}, \tilde{E} = E \cup \{X^j \leftarrow W^j\} \cup \{X^j \leftarrow X_0^j\} \right). \tag{18}$$

By construction, $\mathcal{F}_t(X^i) \subset \mathcal{F}_t(W^i) \vee \mathcal{F}_t(X^{\mathrm{pa}_i^{\mathcal{G}}}) \vee \sigma(X_0^i)$, and therefore

$$\mathcal{F}_t(X^i) \perp\!\!\!\perp \mathcal{F}_t(X^{\mathrm{nd}_i^{\mathcal{G}} \backslash \mathrm{pa}_i} \vee W^{-i} \vee X_0^{-i}) \mid \mathcal{F}_t(X^{\mathrm{pa}_i}) \vee \mathcal{F}_t(W^i) \vee \sigma(X_0^i)$$

where the parent and nondescendant sets are to be read in the graph $\mathcal{G}$, $W^{-i} = \{W_1, \dots, W_n\} \backslash \{W_i\}$, and $X_0^{-i} = \{X_0^1, \dots, X_0^n\} \setminus \{X_0^i\}$. The above statement corresponds to eq. (15) when $v \in \{X_1, \dots, X_n\}$. When $v = W_i$ or $v = X_0^i$, $v$ has no parents in $\tilde{\mathcal{G}}$ and eq. (15) also holds. Therefore, the independence model $\tilde{\mathcal{J}}(P)$ satisfies the local directed Markov property w.r.t. to $\tilde{\mathcal{G}}$.

The independence model $\tilde{\mathcal{J}}(P)$ is a semigraphoid, and by Theorem A.10 it satisfies eq. (14) w.r.t. $\tilde{\mathcal{G}}$. We have $X^i \perp\!\!\!\perp_{d-sep} X^{\mathrm{nd}_i \backslash \mathrm{pa}_i} \mid X^{\mathrm{pa}_i}$ in $\tilde{\mathcal{G}}$, and one therefore has

$$\mathcal{F}_t(X^i) \perp\!\!\!\perp \mathcal{F}_t(X^{\mathrm{nd}_i \backslash \mathrm{pa}_i}) \mid \mathcal{F}_t(X^{\mathrm{pa}^i})$$

establishing eq. (15) for $\mathcal{J}(P)$ w.r.t. $\mathcal{G}$. $\square$

## A.10 PROOF OF COROLLARY 3.3 AND ALGORITHM FOR ROBUST CAUSAL DISCOVERY

*Proof.* This is clear from the data generating mechanism in eq. (1) where all Brownian motions $\mathrm{d}W_t^k$ and initial conditions $X_0^k$ are jointly independent. Since $X_{[0,T]}^j$ is fully determined by $\{X_0^{\mathrm{an}_j}, \mathrm{d}W_t^{\mathrm{an}_j}\}$ where $\mathrm{an}_j$ are the ancestors of $j$ in $\mathcal{G}$, it follows that $i \in \mathrm{an}_j$ if and only if $X_{[0,T]}^j \not\perp\!\!\!\perp X_0^i$. Since we already know from the CPDAG that $i$ and $j$ are adjacent and the graph is acyclic, $i \in \mathrm{an}_j$ implies $i \to j$. □

The full written out algorithm then looks like Algorithm 2 and is sound and complete by Corollary 3.3.

---

**Algorithm 2** Robust Causal discovery for acyclic SDE models leveraging the initial values

---

1: $V_{pre} \leftarrow \{k \in V\}$
  $E_{pre} \leftarrow \{i \to j, \, j \to i \mid i, j \in V, \, i \neq j\}$
2: **for** $c = 0, \ldots, d - 2$ **do** ▷ *Adjacency detection*
3:   **for** $i, j \in V, i \neq j$ **do**
4:    **for** $K \subseteq V \setminus \{i, j\}, |K| = c$,
      s.t. $(k\!-\!j) \in E_{pre}$ for $k \in K$ **do**
5:     **if** $X^i \perp\!\!\!\perp X^j \mid X^K$ **then**
6:      $E_{pre} \leftarrow E_{pre} \setminus \{i \to j, j \to i\}$
       $S_{ij} \leftarrow V$
7: **for** each triple $i, j, k \in V$ with $i\!-\!j\!-\!k$ and $i\!\not\!-\!k$ **do** ▷ *Collider Orientation*
8:   **if** $j \notin S_{ik}$ **then**
9:    $E_{pre} \leftarrow E_{pre} \setminus \{(j, i), (j, k)\}$
10: Apply the Meek-Rules
11: **for** $i, j \in V, i \neq j$ s.t. $(i, j), (j, i) \in E$ **do**
12:   **if** $X_0^i \perp\!\!\!\perp X_{[0,1]}^j$ **then**
13:    $E \leftarrow \setminus \{i \to j\}$
14:   **else**
15:    $E \leftarrow \setminus \{j \to i\}$
16: **for** $k \in V$ **do** ▷ *removing loops*
17:   **if** $X^k \perp\!\!\!\perp_{s,h}^{\circlearrowleft} \mid X^{\mathrm{pa}_k^{\mathcal{G}} \setminus \{k\}}$ **then**
18:    $E \leftarrow E \setminus \{k \to k\}$
19: **return** $\mathcal{G}$

---

## A.11 POST-PROCESSING WITHOUT JOINTLY INDEPENDENT INITIAL VALUES

In real-world settings where there is no natural 'starting point' of the processes, we can typically not assume joint independence of initial values $X_0^i$ as 'the process also happened before'. In this subsection we extend Algorithm 2 to a version that is applicable in scenarios where we cannot assume the initial values $(X_0^i)_{i \in [d]}$ to be jointly independent. The algorithm works similar as to Algorithm 2, except for the fact that for each pair $i, j \in V$ with an unoriented edge, we have to condition on the parents of $j$ when testing for the existence of edge $i \to j$ to prevent information flowing from $i$ to $j$. As it can happen that $j$ is connected to $k \neq i$ with another undirected edge, the set $\mathcal{K}_{i \to j}$ is required:

$$\mathcal{K}_{i \to j} := \mathrm{pa}_{j,known}^{\mathcal{G}_{post}} \times \mathcal{P}(V_{j,un}),$$

where $\mathrm{pa}_{j,known}^{\mathcal{G}_{post}} := \{k \in V \setminus \{i, j\} : (k_0, j_1) \in \tilde{E}_{post}, (j_0, k_1) \notin \tilde{E}_{post}\}$, $V_{j,un} := \{k \in V \setminus \{i\} : (k_0, j_1) \in \tilde{E}_{post}, (j_0, k_1) \in \tilde{E}_{post}\}$. $\mathcal{K}_{i \to j}$ is required to implement the procedure of conditioning on all known parents of j ($\mathrm{pa}_{j,known}^{\mathcal{G}_{post}}$) and in case there are unspecified other neighbors $V_{j,un}$ of $j$, loops through the options of either condition on them or don't condition on them. In case $j$ has no other undirected adjacencies, this set only contains one element, the parents of $j$ except $i$.

**Proposition A.12.** *Algorithm 3 is sound and complete assuming faithfulness, meaning if we denote the output graph of Algorithm 3 by $\mathcal{G}_{ext} = (V, E_{ext})$, the output edges $E_{ext}$ coincide with the edges of the true DAG $\mathcal{G} = (V, E)$.*

---

**Algorithm 3** Robust causal discovery for acyclic SDE models.

---

1: $V_{pre} \leftarrow \{k \in V\}$
   $E_{pre} \leftarrow \{i \to j,\ j \to i \mid i, j \in V,\ i \neq j\}$
2: **for** $c = 0, \ldots, d - 2$ **do**                                                            ▷ *Adjacency detection*
3:      **for** $i, j \in V,\ i \neq j$ **do**
4:          **for** $K \subseteq V \setminus \{i, j\},\ |K| = c,$
                  s.t. $(k \!-\! j) \in E_{pre}$ for $k \in K$ **do**
5:              **if** $X^i \perp\!\!\!\perp X^j \mid X^K$ **then**
6:                  $E_{pre} \leftarrow E_{pre} \setminus \{i \to j, j \to i\}$
                    $S_{ij} \leftarrow V$
7: **for** each triple $i, j, k \in V$ with $i \!-\! j \!-\! k$ and $i \!\not-\! k$ **do**                ▷ *Collider Orientation*
8:      **if** $j \notin S_{ik}$ **then**
9:          $E_{pre} \leftarrow E_{pre} \setminus \{(j, i), (j, k)\}$
10: Apply the Meek-Rules
11: $\tilde{V} \leftarrow V$
   $\tilde{E}_{post} \leftarrow \{i_0 \to j_0,\ i_1 \to j_1, i_0 \to j_1 \mid (i, j) \in E_{pre}\}$
          $\cup\ \{i_0 \to i_1 \mid i \in V\}$                                                   ▷ *Lifted Graph*
12: **for** $i, j \in V,\ i \neq j$ s.t. $(i_0, j_1), (j_0, i_1) \in E$ **do**
13:      **for** $K \in \mathcal{K}_{i \to j}$ **do**
14:          **if** $X^i \perp\!\!\!\perp^+_{s,h} X^j \mid X^K$ **then**
15:              $\tilde{E} \leftarrow \tilde{E} \setminus \{i_0 \to j_0, i_1 \to j_1, i_0 \to j_1\}$
                Break the loop
16: $\mathcal{G} = (V, E) \leftarrow \text{collapse}(\tilde{V}, \tilde{E})$
17: **for** $k \in V$ **do**                                                              ▷ *removing loops*
18:      **if** $X^k \perp\!\!\!\perp^{\circlearrowleft}_{s,h} \mid X^{\text{pa}^{\mathcal{G}}_k \setminus \{k\}}$ **then**
19:          $E \leftarrow E \setminus \{k \to k\}$
20: **return** $\mathcal{G}$

---

*Proof.*

"$\supset$": We have to show $(i, j) \in E \implies (i, j) \in E_{ext}$. This direction is clear but will still be stated here. Since the original PC algorithm (see e.g. Spirtes et al. (2000)) is sound, $(i, j) \in \tilde{E}_{post}$ Since $(i, j) \in E$, $i_0$ and $j_1$ cannot be separated in the lifted graph $\forall K \subset V \setminus \{i, j\}$. Thus by the faithfulness assumption $X^i \not\perp\!\!\!\perp^+_{s,h} X^j \mid X^K \ \forall K \subset V \setminus \{i, j\}$ which holds especially for $K \in \mathcal{K}_{i \to j}$, hence $(i, j) \in E_{ext}$.

"$\subset$": *Proof by contraposition:* $(i, j) \in E_{ext} \implies (i, j) \in E$ is equivalent to $(i, j) \notin E \implies (i, j) \notin E_{ext}$. Since the PC result $\mathcal{G}_{pre} = (V, E_{pre})$ has the correct skeleton and v-structures, the only option would be that wlog $i \neq j$ s.t. $j \to i \in \mathcal{G}$, $(i_0, j_1), (j_0, i_1) \in \tilde{E}_{post}$ meaning the original PC was to detect an adjacency but unable to discard $(i, j)$. But then $j_0$ is separated from $i_1$ by the parents $\text{pa}^{\mathcal{G}}_i$ of $i$, which do not contain $j$. Therefore, they are contained in $\mathcal{K}_{j \to i}$ as a single element and by the global Markov property, we obtain $X^j \perp\!\!\!\perp^+_{s,h} X^i \mid X^{\text{pa}^{\mathcal{G}}_i}$, thus $(i, j) \notin E_{ext}$.

$\square$

## A.12    SENSITIVITY OF CONSTRAINT-BASED METHODS TO ERRORS IN CONDITIONAL INDEPENDENCE TESTING

Constraint-based causal discovery methods such as PC or FCI heavily rely on conditional independence tests to infer the causal structure. However, the adaptive nature of these tests, where the results of subsequent tests are contingent upon the outcomes of earlier ones, complicates the error analysis of these approaches. While this is a widely acknowledged challenge faced by all constraint-based methods, how the final graph depends on type I and II errors remains an important open problem in the field. Already early work by Spirtes & Meek (1995) presented a simulation study showing the sensitivity of different methods to the different types of errors, underscoring a variable trade-off between sensitivity and specificity across different sample sizes in skeleton and edge orientation detection depending on the size of the graph and number of samples. Instead, Kalisch & Bühlman

(2007) provide theoretical results showing that the PC algorithm is uniformly consistent for high-dimensional settings under a mild sparsity assumption on the DAGs, where the number of nodes can grow quickly with the sample size. However, they critically rely on the restrictive assumption that the observational distribution is a multi-variate Gaussian for their finite sample results, where a strong theory for testing CI with partial correlations is available.

The performance of conditional independence tests is critical for the robustness of constraint-based methods and consistency (as demonstrated for our test in Appendix A.14) is a critical first step. In practice, extensive simulation studies are another useful approach to benchmark the actual performance of the final causal discovery method.

### A.13   I.I.D. ASSUMPTION OF THE CONDITIONAL INDEPENDENCE TEST

KCIPT and SDCIT both require i.i.d. samples as input. In our analysis, a solution path of all variables in $X$ is treated as a single example originating from the SDE model. By generating $n$ solution paths as independent realizations of a stochastic process (e.g., independent initial conditions and independent Brownian motions in the SDE), we obtain $n$ i.i.d. samples. It is important to note that while multiple observations from each path at different time points are considered part of a single sample, these within-path observations are not assumed to be i.i.d. However, as each path is an independent realization, the collection of different paths meets the i.i.d. requirement necessary for the effective application of KCIPT and SDCIT in our setting.

### A.14   CONSISTENCY OF THE CONDITIONAL INDEPENDENCE TEST

In this section, we argue, that under certain assumptions on our system, Theorem 1 of Doran et al. (2014) still holds even without assuming the existence of a density. At first, it is to be noted that the signature kernel eq. (3) is bounded on compact sets $C \subset \mathrm{BV}(C(I, \mathbb{R}^{d_x}))$ of BV paths, an assumption that can be made as we evaluate our SDEs over finite, discrete time steps. Satisfying this boundedness, it defines an RKHS over these compact sets. In addition, the signature kernel is characteristic on these bounded sets (see Proposition 3.3 Cass et al. (2023)). We use the notation $X : (\Omega, \mathcal{A}, P) \to (\mathcal{X}, \mathcal{A}_{\mathcal{X}})$, $Y : (\Omega, \mathcal{A}, P) \to (\mathcal{Y}, \mathcal{A}_{\mathcal{Y}})$, $Z : (\Omega, \mathcal{A}, P) \to (\mathcal{Z}, \mathcal{A}_{\mathcal{Z}})$ with $\mathcal{X}, \mathcal{Y}, \mathcal{Z}$ being compact subsets in $\mathrm{BV}(C(I, \mathbb{R}^{n_x})), \mathrm{BV}(C(I, \mathbb{R}^{n_y})), \mathrm{BV}(C(I, \mathbb{R}^{n_z}))$ and denote the RKHS's defined by the signature kernel as $(\mathcal{H}_{\mathcal{X}}, k_{\mathcal{X}}), (\mathcal{H}_{\mathcal{Y}}, k_{\mathcal{Y}}), (\mathcal{H}_{\mathcal{Z}}, k_{\mathcal{Z}})$.

Our measure-theoretical argumentation now follows the definitions in Park & Muandet (2020) and the original proof in Doran et al. (2014). Let $H$ be a Banach space, $\mathcal{A}'_H \subset \mathcal{A}_H$ a sub-$\sigma$-algebra of the Borel-$\sigma$-algebra.

**Definition A.13.** Let $F : (\Omega, \mathcal{A}, P) \to (H, \mathcal{A}_H)$ a Bochner $P$-integrable random variable. The *conditional expectation of $H$* w.r.t. $E$ is a Bochner $P$-integrable RV $F : (\Omega, \mathcal{A}, P) \to (H, \mathcal{A}'_H)$ s.t.

$$\forall A \in \mathcal{A}'_H \quad \int_A F \mathrm{d}P = \int_A F' \mathrm{d}P \,.$$

Due to the existence and (almost sure) uniqueness, such an $F'$ is denoted $\mathbb{E}[F \mid \mathcal{A}'_H]$ with the notation $\mathbb{E}[F \mid Z] := \mathbb{E}[F \mid \sigma(Z)]$ for a general $Z : (\Omega, \mathcal{A}, P) \to (\mathcal{Z}, \mathcal{A}_{\mathcal{Z}})$ which in turn can be used to define the *conditional expectation* $P(A \mid \mathcal{A}'_H) := \mathbb{E}[\mathbb{1}_A \mid \mathcal{A}'_H]$ and can be used to define the *conditional kernel mean embedding*

$$\mu_{X|Z} := \mu_{P_{X|Z}} := \mathbb{E}_{X|Z}[k_{\mathcal{X}}(X, \cdot) \mid Z]$$

As proven in Theorem 5.8 of Park & Muandet (2020) under the assumption, that $P_{X|Z} := \mathbb{E}[\cdot, \mid Z]$ admits a regular version (meaning its can be written as a transition kernel, see Park & Muandet (2020)) and $k_{\mathcal{X}} \otimes k_{\mathcal{Y}}$ is characteristic, it holds that

$$X \perp\!\!\!\perp Y \mid Z \Leftrightarrow \|\mu_{XY|Z} - \mu_{X|Z} \otimes \mu_{Y|Z}\|_{\mathcal{H}_{\mathcal{X}} \otimes \mathcal{H}_{\mathcal{Y}}} = 0 \Leftrightarrow \|\mu_{XY|Z} \otimes \mu_Z - \mu_{X|Z} \otimes \mu_{Y|Z} \otimes \mu_Z\|_{\mathcal{H}_{\mathcal{X}} \otimes \mathcal{H}_{\mathcal{Y}} \otimes \mathcal{H}_{\mathcal{Z}}} = 0$$

where the second equivalence is trivial. Let now $\sigma \in \mathcal{S}_n$ be a permutation such that its permutation matrix $M_\sigma = (m_{ij})$

$$m_{ij} = \begin{cases} 1 & \text{if } j = \sigma(i) \,, \\ 0 & \text{if } j \neq \sigma(i) \,. \end{cases}$$

satisfies $\mathrm{Tr}(M_\sigma) = 0$. The empirical estimates for the given sample $\{(x_i, y_i, z_i)\}_{i \in [n]}$ and its permuted version under $\sigma$

$$\hat{\mu}_{P_{X,Y,Z}} := \frac{1}{n} \sum_{i=1}^{n} k_{\mathcal{X}}(x_i, \cdot) \otimes k_{\mathcal{Y}}(y_i, \cdot) \otimes k_{\mathcal{Z}}(z_i, \cdot), \tag{19}$$

$$\hat{\mu}_{P'_{X,Y,Z}} := \frac{1}{n} \sum_{i=1}^{n} k_{\mathcal{X}}(x_i, \cdot) \otimes k_{\mathcal{Y}}(y_{\sigma(i)}, \cdot) \otimes k_{\mathcal{Z}}(z_i, \cdot). \tag{20}$$

Similar to Laumann et al. (2023), can now write the distance between test statistic and the approximated test statistic in terms of the sum of the distance between the empirical estimates and its respective counterparts by using the inverse and the regular triangle inequality

$$\left| \|\mu_{XY|Z} \otimes \mu_Z - \mu_{X|Z} \otimes \mu_{Y|Z} \otimes \mu_Z\|_{\mathcal{H}_{\mathcal{X}} \otimes \mathcal{H}_{\mathcal{Y}} \otimes \mathcal{H}_{\mathcal{Z}}} - \|\hat{\mu}_{P_{X,Y,Z}} - \hat{\mu}_{P'_{X,Y,Z}}\|_{\mathcal{H}_{\mathcal{X}} \otimes \mathcal{H}_{\mathcal{Y}} \otimes \mathcal{H}_{\mathcal{Z}}} \right|$$

$$\leq \left| \|\mu_{XY|Z} \otimes \mu_Z - \mu_{X|Z} \otimes \mu_{Y|Z} \otimes \mu_Z - \hat{\mu}_{P_{X,Y,Z}} + \hat{\mu}_{P'_{X,Y,Z}}\|_{\mathcal{H}_{\mathcal{X}} \otimes \mathcal{H}_{\mathcal{Y}} \otimes \mathcal{H}_{\mathcal{Z}}} \right|$$

$$\leq \|\mu_{XY|Z} \otimes \mu_Z - \hat{\mu}_{P_{X,Y,Z}}\|_{\mathcal{H}_{\mathcal{X}} \otimes \mathcal{H}_{\mathcal{Y}} \otimes \mathcal{H}_{\mathcal{Z}}} + \|\mu_{X|Z} \otimes \mu_{Y|Z} \otimes \mu_Z - \hat{\mu}_{P'_{X,Y,Z}}\|_{\mathcal{H}_{\mathcal{X}} \otimes \mathcal{H}_{\mathcal{Y}} \otimes \mathcal{H}_{\mathcal{Z}}}.$$

By Park & Muandet (2020), the first term tends to zero for $n \to \infty$. With a similar line of argumentation as in Doran et al. (2014) on can establish a majorant for the estimated HSCIC, namely

$$\|\hat{\mu}_{P_{X,Y,Z}} - \hat{\mu}_{P'_{X,Y,Z}}\|_{\mathcal{H}_{\mathcal{X}} \otimes \mathcal{H}_{\mathcal{Y}} \otimes \mathcal{H}_{\mathcal{Z}}}$$

$$= \left\| \frac{1}{n} \sum_{i=1}^{n} k_{\mathcal{X}}(x_i, \cdot) \otimes k_{\mathcal{Y}}(y_i, \cdot) \otimes k_{\mathcal{Z}}(z_i, \cdot) - \frac{1}{n} \sum_{i=1}^{n} k_{\mathcal{X}}(x_i, \cdot) \otimes k_{\mathcal{Y}}(y_{\sigma(i)}, \cdot) \otimes k_{\mathcal{Z}}(z_i, \cdot) \right\|_{\mathcal{H}_{\mathcal{X}} \otimes \mathcal{H}_{\mathcal{Y}} \otimes \mathcal{H}_{\mathcal{Z}}}$$

$$= \left\| \frac{1}{n} \sum_{i=1}^{n} k_{\mathcal{X}}(x_i, \cdot) \otimes k_{\mathcal{Y}}(y_i, \cdot) \otimes k_{\mathcal{Z}}(z_i, \cdot) - \frac{1}{n} \sum_{i=\sigma(j), j=1}^{n} k_{\mathcal{X}}(x_{\sigma^{-1}(i)}, \cdot) \otimes k_{\mathcal{Y}}(y_i, \cdot) \otimes k_{\mathcal{Z}}(z_{\sigma^{-1}(i)}, \cdot) \right\|_{\mathcal{H}_{\mathcal{X}} \otimes \mathcal{H}_{\mathcal{Y}} \otimes \mathcal{H}_{\mathcal{Z}}}$$

$$\leq \frac{1}{n} \sum_{i=1}^{n} \left\| \left( k_{\mathcal{X}}(x_i, \cdot) - k_{\mathcal{X}}(x_{\sigma^{-1}(i)}, \cdot) \right) \otimes k_{\mathcal{Y}}(y_i, \cdot) \otimes \left( k_{\mathcal{Z}}(z_i, \cdot) - k_{\mathcal{Z}}(z_{\sigma^{-1}(i)}, \cdot) \right) \right\|_{\mathcal{H}_{\mathcal{X}} \otimes \mathcal{H}_{\mathcal{Y}} \otimes \mathcal{H}_{\mathcal{Z}}}$$

$$\leq \frac{1}{n} \sum_{i=1}^{n} \underbrace{\left\| k_{\mathcal{X}}(x_i, \cdot) - k_{\mathcal{X}}(x_{\sigma^{-1}(i)}, \cdot) \right\|_{\mathcal{H}_{\mathcal{X}}}}_{\leq 2M_x} \underbrace{\left\| k_{\mathcal{Y}}(y_i, \cdot) \right\|_{\mathcal{H}_{\mathcal{Y}}}}_{\leq M_y} \left\| \left( k_{\mathcal{Z}}(z_i, \cdot) - k_{\mathcal{Z}}(z_{\sigma^{-1}(i)}, \cdot) \right) \right\|_{\mathcal{H}_{\mathcal{Z}}}$$

$$\leq 2M_x M_y \left( \frac{1}{n} \mathrm{Tr}(PD^{RKHS}) \right)$$

where the assumptions can be made due to the boundedness of the kernels. Hence, under the assumption that

$$\frac{1}{n} \sum_{i=1}^{n} \delta_{(x_i, y_{\sigma(i)}, z_i)} \to P_{X|Z} \otimes P_{Y|Z} \otimes P_Z \tag{21}$$

a statement that holds true in the case of densities (see supplementary material of Janzing et al. (2013)), since $P_{\cdot|Z}$ is regular, $\hat{\mu}_{P'_{X,Y,Z}} \to \mu_{X|Z} \otimes \mu_{Y|Z} \otimes \mu_Z$.

Hence under the assumptions that $P_{\cdot|Z}$ admits a regular version (an assumption also made by Laumann et al. (2023)), the considered random variables operate on compact sets of BV-paths and eq. (21) holds, the test is consistent if and only if $\frac{1}{n} \mathrm{Tr}(PD^{RKHS}) \to 0$. This is the exact same statement as in Theorem 1 of Doran et al. (2014).

## B  EXPERIMENTS AND EVALUATION

This section provides details about the experimental framework, elaborating on the data generation process, the parameters and architectures employed, and presents additional results.

Table 5: Performance of *future-extended h-locally conditionally independence criterion* in the bivariate setting for the two heuristic $\sigma_1$ and $\sigma_2$ for different sample sizes. The SHD is measured for different values of $s$ in $\perp\!\!\!\perp^+_{s,t}$ with $t = 1$, defining different lengths of intervals in the past and future. Overall the performance is best of shorter values of the past intervals compared to the future ($s < \frac{t}{2}$).

| $(\times 10^2)$ | $\sigma_1$ | | $\sigma_2$ | |
|---|---|---|---|---|
| | $n = 400$ | $n = 600$ | $n = 400$ | $n = 600$ |
| $s = 0.1$ | $\mathbf{76.2 \pm 2.5}$ | $\mathbf{58.4 \pm 2.7}$ | $14.9 \pm 2.0$ | $15.8 \pm 2.0$ |
| $s = 0.2$ | $77.3 \pm 2.7$ | $72.2 \pm 2.8$ | $\mathbf{12.9 \pm 1.7}$ | $\mathbf{10.9 \pm 1.7}$ |
| $s = 0.3$ | $80.2 \pm 2.3$ | $76.2 \pm 2.6$ | $21.8 \pm 2.2$ | $17.8 \pm 2.0$ |
| $s = 0.4$ | $89.1 \pm 2.0$ | $92.0 \pm 2.1$ | $31.7 \pm 2.4$ | $25.7 \pm 2.2$ |
| $s = 0.5$ | $89.1 \pm 2.0$ | $98.1 \pm 2.0$ | $38.6 \pm 2.5$ | $27.7 \pm 2.2$ |
| $s = 0.6$ | $82.4 \pm 1.9$ | $96.0 \pm 2.1$ | $60.0 \pm 2.7$ | $48.5 \pm 2.9$ |
| $s = 0.7$ | $102.0 \pm 1.2$ | $104.0 \pm 2.0$ | $80.2 \pm 2.2$ | $61.4 \pm 2.7$ |
| $s = 0.8$ | $104.0 \pm 1.4$ | $95.0 \pm 1.5$ | $81.1 \pm 2.0$ | $87.1 \pm 2.4$ |
| $s = 0.9$ | $95.0 \pm 1.6$ | $104.4 \pm 1.8$ | $102.0 \pm 1.2$ | $101.0 \pm 2.1$ |

## B.1 KERNEL HEURISTIC AND INTERVAL CHOICE

In kernel methods, besides selecting the right kernel function, the correct choice of its parameters is vital. A reasonable choice for radially symmetric kernels $k(x_i, x_j) = f(\|x_i - x_j\|/\sigma)$, $f : \mathbb{R}_+ \to \mathbb{R}_+$ is to choose $\sigma$ to lie in the same order of magnitude as the pairwise distances $(\|x_i - x_j\|)_{i,j}$. Choosing the bandwidth for the RBF kernel in our setting with samples $(x_i)_{i \in [n]}$, $x_i \in \mathcal{C}^0(I, \mathbb{R}^d)$ is not straight forward, as the RBF kernel ultimately acts on signatures and there is no immediate way in obtaining a 'typical scale' of pairwise distances of signatures. To still set the bandwidth purely based on observed data, there are two natural ways to implement a median heuristic based on two different collections of pairwise distances. The first option is to choose

$$\sigma_1 = \mathrm{Median}\{\|x_i - x_j\|_{L^2} \mid i, j \in [n]\},$$

where $\|\cdot\|_{L^2}$ is the is the $L^2$ norm of entire paths, i.e., integrated over the time interval $I$. Given that in practice we observe the paths $x_i$ still at fixed time points $t_1 < \ldots < t_m$ within the interval $I$, we can also consider the heuristic

$$\sigma_2 = \mathrm{Median}\{\|x_{i,t_l} - x_{j,t_k}\|_2 \mid i, j \in [n], l, k \in [m]\},$$

where $\|\cdot\|_2$ is the Euclidean norm in $\mathbb{R}^d$ and we consider pairwise distances of all observations across all paths and time points. In early benchmarks, we have empirically found $\sigma_2$ to yield a better heuristic by comparing the performance of both heuristics by evaluation of the SHD in a causal discovery task in the linear bivariate setting $X^1 \to X^2$ without diffusion interaction with $a_{21} \sim \mathcal{U}((-2.5, -1] \cup [1, 2.5))$, $a_{11}, a_{22} \sim \mathcal{U}((-0.5, 0.5))$ and $\sigma_i \sim \mathcal{U}([0, 0.5))$. The experiment is performed for different choices of intervals $[0, s]$, $[s, s+h]$ in the future-extended $h$-local conditional independence. As can be seen from Table 5, $\sigma_2$ and the intervals $[0, 0.1], [0.1, 1]$ seem to perform best on average and are therefore used in the rest of the experiments. In all the experiments of the paper, in the implementation of the signature kernel we use a depth parameter of 4, the RBF kernel and we add time as an extra dimension to the signature kernel.

## B.2 EMPIRICAL COMPARISON OF DIFFERENT TESTS

Different kernel-based conditional and unconditional tests are compared. HSIC with bootstrap (HSIC$_b$) and with $\gamma$-distribution approximation (HSIC$_\gamma$) (Gretton et al., 2007) were implemented for the unconditional setting, while KCIPT (Doran et al., 2014), KCIT with bootstrap (KCIT$_b$), KCIT with $\gamma$ approximation (KCIT$_\gamma$) (Zhang et al., 2011) and SDCIT for the conditional setting (Lee & Honavar, 2017). Figure 6 shows that SDCIT and HSIC$_b$ are overall the best performing ones in terms of type I and type II error. In this experiment, we sampled 100 different linear SDEs and we sample 400 different trajectories within time interval [0,1]. The number of bootstrap samples over permutation test is set to 100, the number of permutations for a single permutation test to 20000, the

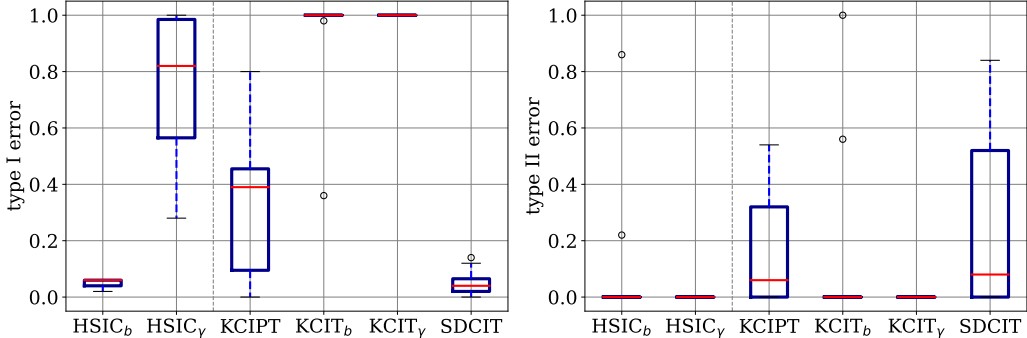

Figure 6: Performance in terms of type I and type II errors of different unconditional ($HSIC_b$ and $HSIC_\gamma$) and conditional tests (KCIPT, $KCIT_b$, $KCIT_\gamma$, SDCIT). Among the conditional tests, SDCIT outperforms KCIPT in terms of type I error, while performing in a similar range in terms of type II error. Among the unconditional tests, $HSIC_b$ outperforms $HSIC_\gamma$

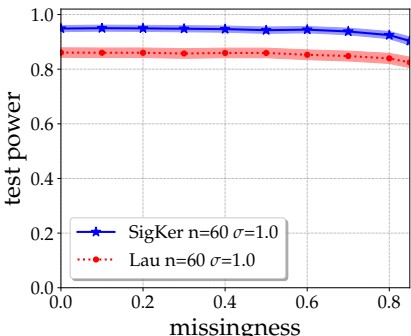

Figure 7: Test power over the missingness level (percentage of observations that is being dropped at random). While both our method as well as Laumann et al. (2023) maintain their power up to high levels of missingness, our test still consistently outperforms the baseline. Lines show means over 1000 settings in the power analysis in Section 4 and shaded regions show standard errors.

number of null samples via Monte Carlo from all values in permutation test to 20000 and 1000 for HSIC, the number of null samples for KCIT-bootstrap to 20000 and the number of null samples for SDCIT is set to 1000. These are the tests parameters also used in the rest of the experiments.

### B.3 ADDITIONAL EXPERIMENTAL RESULTS FOR THE BIVARIATE CASE

Figure 7 shows that in the simple unconditional bivariate setting of the power analysis in Section 4 our test maintains high power up to substantial levels of missingness and only starts degrading once more than 80% of observations are dropped at random starting from 64 time steps over the interval $[0, 1]$ for 1000 different SDEs.

### B.4 BUILDING BLOCKS FOR CAUSAL DISCOVERY

The performance of the conditional and unconditional independence tests was evaluated using widely recognized causal structures that are at the foundation of causal discovery. Specifically, we examined scenarios involving two variables, both connected and unconnected, and three-variable configurations comprising chain, fork, and collider structures. The analysis of type I and type II errors, as shown in Table 6, indicates robust performance for both conditional SDCIT and unconditional HSIC tests.

Table 6: Test performance on the key building blocks for causal discovery. Here, $\perp\!\!\!\perp$ refers to $\perp\!\!\!\perp_{\text{sym}}$ except for $X_p^1 \perp\!\!\!\perp X_f^1$ which means $X_{[0,t/2]}^1 \perp\!\!\!\perp X_{[t/2,t]}^1$. We use SDCIT for CI tests and bootstrapped HSIC unconditional independence. We simulate 40 samples of 512 trajectories for 10 different SDEs (400 tests per column) with 25% of the 128 original observations dropped uniformly at random. The **error** represent type I or type II error when the null hypothesis $H_0$ should be rejected (✗) or accepted (✓), respectively.

| graph | $X^1 \to X^2 \to X^3$ | $X^1 \to X^2 \leftarrow X^3$ | $X^1 \leftarrow X^2 \to X^3$ | $X^1$ | $X^1 \quad X^2$ | $X^1 \to X^2$ |
|---|---|---|---|---|---|---|
| $H_0$ | $X^1 \perp\!\!\!\perp X^2\|X^3$ $\quad$ $X^1 \perp\!\!\!\perp X^3\|X^2$ $\quad$ $X^1 \perp\!\!\!\perp X^2\|X^3$ $\quad$ $X^1 \perp\!\!\!\perp X^3\|X^2$ | | | $X^1 \perp\!\!\!\perp X^3\|X^2$ | $X_p^1 \perp\!\!\!\perp X_f^1$ $\quad$ $X^1 \perp\!\!\!\perp X^2$ | $X^1 \perp\!\!\!\perp X^2$ |
| should reject | ✓ $\quad$ ✗ $\quad$ ✓ $\quad$ ✓ | | | ✗ | ✓ $\quad$ ✗ | ✓ |
| **error** | 0.0000 $\quad$ 0.0075 $\quad$ 0.0000 $\quad$ 0.0000 | | | 0.9600 | 0.0000 $\quad$ 0.0450 | 0.0000 |

Table 7: Test performance on the key building blocks for causal discovery. Here, $\perp\!\!\!\perp$ refers to $\perp\!\!\!\perp_{\text{sym}}$ except for $X_p^1 \perp\!\!\!\perp X_f^1$ which means $X_{[0,t/2]}^1 \perp\!\!\!\perp X_{[t/2,t]}^1$ in a setting where the dependence only comes from the diffusion. We use SDCIT for CI tests and bootstrapped HSIC unconditional independence. We simulate 40 samples of 512 trajectories for 10 different SDEs (400 tests per column) with 25% of the 128 original observations dropped uniformly at random. The **error** represent type I or type II error when the null hypothesis $H_0$ should be rejected (✗) or accepted (✓), respectively.

| graph | $X^1 \to X^2 \to X^3$ | $X^1 \to X^2 \leftarrow X^3$ | $X^1 \leftarrow X^2 \to X^3$ | $X^1$ $\quad$ $X^1$ | $X^2$ | $X^1 \to X^2$ |
|---|---|---|---|---|---|---|
| $H_0$ | $X^1 \perp\!\!\!\perp X^2\|X^3$ $\quad$ $X^1 \perp\!\!\!\perp X^3\|X^2$ $\quad$ $X^1 \perp\!\!\!\perp X^2\|X^3$ $\quad$ $X^1 \perp\!\!\!\perp X^3\|X^2$ | | | $X^1 \perp\!\!\!\perp X^3\|X^2$ | $X_p^1 \perp\!\!\!\perp X_f^1$ $\quad$ $X^1 \perp\!\!\!\perp X^2$ | $X^1 \perp\!\!\!\perp X^2$ |
| should reject | ✓ $\quad$ ✗ $\quad$ ✓ $\quad$ ✓ | | | ✗ | ✓ $\quad$ ✗ | ✓ |
| **error** | 0.0000 $\quad$ 0.0072 $\quad$ 0.0000 $\quad$ 0.0000 | | | 0.9600 | 0.00 $\quad$ 0.07 | 0.01 |

These results refer to the symmetric conditional independence $\perp\!\!\!\perp_{sym}$, while future-extended h-local conditional independence are presented in Table 8. The tables show that the test is able to reconstruct the presence and the directionality of the edges overall all building blocks of causal structure learning. We also perform the same experiments for the setting in which the causal dependence only comes from the diffusion. Referring to the linear SDE in eq. (5), $a_{ij} = a_{ji} = 0$, $a_{11}, a_{22} \sim \mathcal{U}[-0.5, 0.5]$, $\sigma_{12}, \sigma_{21} \sim \mathcal{U}[(-2.5, -1.0) \cup (1.0, 2.5)]$ and $\sigma_{11}, \sigma_{22} \sim \mathcal{U}[-0.5, 0.5]$. $b_i$ and $d_i$ are sampled from $\mathcal{U}[-0.1, 0.1]$ and $\mathcal{U}[-0.2, 0.2]$. Table 7 includes the results for all the different causal structures.

## B.5 CAUSAL DISCOVERY

We tested SCOTCH using various sparsity parameters and epochs to identify the optimal configuration. Table 9 confirms that the configuration with $\lambda = 200$ and $n_e = 2000$ outperforms others in the tested settings, and hence is chosen for the comparison with our method in Table 2. The table also reveals that SCOTCH's performance is not robust to changes in $\lambda$, showing substantial variations under the same number of epochs, optimizer, and learning rate with different $\lambda$ values. These results demonstrate that the SCOTCH's performance can significantly deteriorate with changes in the parameter $\lambda$, while keeping the number of epochs constant. For example, when $\lambda$ is adjusted from 200 to 100, the Structural Hamming Distance (SHD, scaled by $10^2$) increases from an average of 538 to 10,275 for $n_e = 2000$.

## B.6 CONDITIONAL INDEPENDENCE TESTING BEYOND THE SDE MODEL

In this section, we aim to demonstrate that our proposed non-parametric conditional independence (CI) test extends beyond the setting of SDEs and even beyond semi-martingales, rendering it more broadly applicable than most existing methods, which usually assume independent (additive) noise.

Table 8: Test performance on the chain, fork and collider causal structures for the *future-extended h-locally conditionally independence criterion*. Here, $\perp\!\!\!\perp$ refers to $\perp\!\!\!\perp_{s,h}^{+}$ and we use SDCIT. We simulate 10 samples of 512 trajectories for 10 different SDEs (100 tests per column). The **error** represent type I or type II error when the null hypothesis $H_0$ should be rejected (✗) or accepted (✓), respectively.

| graph | $X^1 \to X^2 \to X^3$ | | | | | |
|---|---|---|---|---|---|---|
| $H_0$ | $X^1 \perp\!\!\!\perp X^2 \vert X^3$ | $X^2 \perp\!\!\!\perp X^1 \vert X^3$ | $X^1 \perp\!\!\!\perp X^3 \vert X^2$ | $X^3 \perp\!\!\!\perp X^1 \vert X^2$ | $X^2 \perp\!\!\!\perp X^3 \vert X^1$ | $X^3 \perp\!\!\!\perp X^2 \vert X^1$ |
| should reject | ✗ | ✓ | ✗ | ✗ | ✗ | ✓ |
| **error** | 0.07 | 0.33 | 0.04 | 0.04 | 0.03 | 0.00 |
| graph | $X^1 \leftarrow X^2 \to X^3$ | | | | | |
| $H_0$ | $X^1 \perp\!\!\!\perp X^2 \vert X^3$ | $X^2 \perp\!\!\!\perp X^1 \vert X^3$ | $X^1 \perp\!\!\!\perp X^3 \vert X^2$ | $X^3 \perp\!\!\!\perp X^1 \vert X^2$ | $X^2 \perp\!\!\!\perp X^3 \vert X^1$ | $X^3 \perp\!\!\!\perp X^2 \vert X^1$ |
| should reject | ✓ | ✗ | ✗ | ✗ | ✗ | ✓ |
| **error** | 0.12 | 0.02 | 0.04 | 0.02 | 0.04 | 0.06 |
| graph | $X^1 \to X^2 \leftarrow X^3$ | | | | | |
| $H_0$ | $X^1 \perp\!\!\!\perp X^2 \vert X^3$ | $X^2 \perp\!\!\!\perp X^1 \vert X^3$ | $X^1 \perp\!\!\!\perp X^3 \vert X^2$ | $X^3 \perp\!\!\!\perp X^1 \vert X^2$ | $X^2 \perp\!\!\!\perp X^3 \vert X^1$ | $X^3 \perp\!\!\!\perp X^2 \vert X^1$ |
| should reject | ✗ | ✓ | ✓ | ✓ | ✓ | ✗ |
| **error** | 0.08 | 0.4 | 0.56 | 0.46 | 0.36 | 0.16 |

Table 9: SHD ($\times 10^2$) of discovered graphs, for 200 samples and $d$ nodes in the SDE model. The mean and standard error of SCOTCH for different values of $\lambda$ and $n_e$ for 20 different SDEs is presented. The best performing setting for higher dimensional graphs is $\lambda = 200$ and $n_e = 2k$.

| $\times 10^2$ | $(\lambda, n_e)$ | $d = 3$ | $d = 5$ | $d = 10$ | $d = 20$ | $d = 50$ |
|---|---|---|---|---|---|---|
| SCOTCH | 100, 2k | $188 \pm 28$ | $417 \pm 86$ | $250 \pm 61$ | $1525 \pm 1160$ | $10275 \pm 6176$ |
| SCOTCH | 200, 2k | $110 \pm 21$ | $270 \pm 48$ | $530 \pm 223$ | $\mathbf{370 \pm 174}$ | $\mathbf{538 \pm 70}$ |
| SCOTCH | 1, 1k | $\mathbf{20 \pm 11}$ | $\mathbf{85 \pm 20}$ | $\mathbf{375 \pm 28}$ | $1455 \pm 103$ | $6095 \pm 153$ |
| SCOTCH | 50, 1k | $330 \pm 27$ | $1255 \pm 54$ | $5895 \pm 107$ | $1340 \pm 243$ | $2994 \pm 941$ |
| SCOTCH | 200, 1k | $400 \pm 17$ | $1370 \pm 29$ | $6425 \pm 80$ | $705 \pm 77$ | $7863 \pm 2670$ |

**Fractional Brownian Motion.**   To illustrate this, we start by applying our test to a stochastic process driven by fractional Brownian motions (fBMs). Fractional Brownian motion is a generalization of standard Brownian motion, a continuous time Gaussian process $B_H(t)$ with zero mean and $\mathbb{E}[B_H(t)B_H(s)] = \frac{1}{2}\left(|t|^{2H} + |s|^{2H} - |t - s|^{2H}\right)$ that incorporates a parameter $H \in (0, 1)$, the so called *Hurst parameter*, which governs long-range dependencies. For

- $H = 0.5$ it reduces to the regular Brownian Motion.
- $H > 0.5$ increments are positively correlated.
- $H < 0.5$ increments are negatively correlated.

Specifically, we consider the following system of stochastic differential equations driven by fractional Brownian motions

$$\begin{pmatrix} dX_t \\ dY_t \end{pmatrix} = \begin{pmatrix} 0 & 0 \\ a_{21} & 0 \end{pmatrix} \begin{pmatrix} X_t \\ Y_t \end{pmatrix} dt + \begin{pmatrix} d_1 & 0 \\ 0 & d_2 \end{pmatrix} \begin{pmatrix} dW_t^1 \\ dW_t^2 \end{pmatrix}, \tag{22}$$

with $W_t^1, W_t^2$ independent fBMs with Hurst parameter $H$ and $a_{21}, d_1, d_2 \sim \mathcal{U}([-2, 2])$. To quantify the effectiveness of our test in this context, we measure the test power for different strengths of interaction for different Hurst parameters. Figure 8 shows this power analysis for 500 random settings of eq. (22). The power of our test steadily approaches 1 as the interaction strength increases to 1 regardless of $H$. We note that since the above example does not have independent increments, Algorithm 1 is not applicable anymore for causal discovery in such settings. Instead, the fractional Brownian motion example merely highlights the usefulness of our CI test independently from its application in causal discovery in the SDE model.

**Functional data.**   For further corroboration of our claims, we turn to the following example from functional data analysis (FDA), taken from Laumann et al. (2023), to test causal discovery in a non SDE setting. After drawing a DAG $\mathcal{G} = (V, E)$, processes for source vertices, i.e., vertices $i = v_i$ without parents in $\mathcal{G}$, are generated according to

$$X^i(t) = \sum_{m=1}^{M} c_m^i \phi_m(t) + \epsilon_t^i$$

with Fourier basis functions $\phi_1(t) = 1$, $\phi_2(t) = \sqrt{2}\sin(2\pi t)$, $\phi_3(t) = \sqrt{2}\cos(2\pi t)$ and so on, the weights $c_m^i \sim \mathcal{N}(0, 1)$, and the additive noise $\epsilon_t^i \sim \mathcal{N}(0, 1)$. For all vertices $i$ with at least one parent, we set

$$X^i(t) = a \sum_{k \in \mathrm{pa}_i^{\mathcal{G}}} \int_0^t X^k(s)\beta^k(s, t)ds + \epsilon_t^i$$

with $\beta^k(s, t) = 8(s - c_1^k)^2 - 8(t - c_2^k))^2$, $c_1^k, c_2^k \sim \mathcal{U}_{[0,1]}$ and $a = 1$.

We draw 40 DAGs with 40 distinct FD generating mechanisms for each dimension $d \in \{3, 5, 10, 20, 50\}$. Table 10 shows that SCOTCH cannot capture the underlying dependencies, as it is tailored specifically to the SDE model. It is thus misspecified for such a function generating mechanism. Under such (arguably mild) misspecification, SCOTCH is even outperformed by PCMCI, whereas our method still performs much better than both baselines.

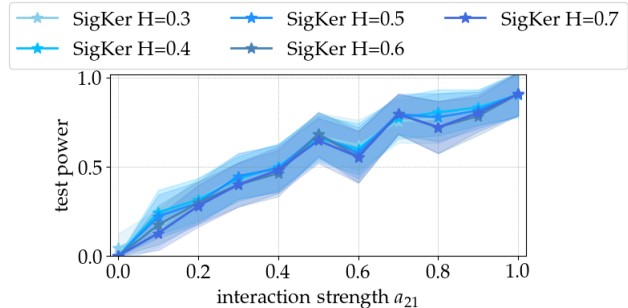

Figure 8: Test power over for different fBM-driven processes eq. (22) for different H. The CI test picks up dependence for as little as 100 samples given a certain amount of interaction-strength. Lines show means over 500 settings and shaded regions show standard errors.

Table 10: SHD ($\times 10^2$) comparison of SigKer to the baselines PCMCI and SCOTCH (with $\lambda = 200$ and $n_e = 2000$) in the functional data setting. We report mean and standard errors over 40 instances.

| $\times 10^2$ | $d = 3$ | $d = 5$ | $d = 10$ | $d = 20$ | $d = 50$ |
|---|---|---|---|---|---|
| $\perp\!\!\!\perp_{\text{sym}}$ | $40 \pm 8$ | $22 \pm 10$ | $509 \pm 25$ | $1542 \pm 60$ | $6786 \pm 223$ |
| $\perp\!\!\!\perp_{\text{sym}}$+p.p. | $42 \pm 8$ | $141 \pm 13$ | $489 \pm 28$ | $1498 \pm 56$ | $6529 \pm 250$ |
| PCMCI | $110 \pm 30$ | $320 \pm 42$ | $1150 \pm 121$ | $6518 \pm 293$ | $7799 \pm 250$ |
| SCOTCH | $558 \pm 14$ | $1861 \pm 43$ | $8567 \pm 63$ | $35467 \pm 833$ | $105165 \pm 10934$ |

## B.7 REAL-WORLD PAIRS TRADING EXAMPLE

In the pairs trading experiment, stock price data is downloaded from Yahoo Finance for a predefined list of stocks over a specific period, divided into training (1st January 2010 to 31st December 2011) and trading intervals (1st January 2012 to 31st December 2012). The chose stocks are Trinity Industries (TRN), Brandywine Realty Trust (BDN), Commercial Metals Company (CMC), The New York Times Company (NYT), New York Community Bancorp (NYCB), The Wendy's Company (WEN), CNX Resources Corporation (CNX). Logarithms of the stock prices are computed to stabilize variance and normalize the prices. During the training period, pairs of stocks are selected based on various statistical tests: Cointegration Test (Engle-Granger) to determine if a pair of stocks is cointegrated, suggesting a long-term equilibrium relationship; Augmented Dickey-Fuller (ADF) Test to assess the stationarity of the spread (ratio) of a pair's prices; and Granger Causality Test to check if the price of one stock in the pair can predict the price of the other. Pairs are selected based on the p-values from these tests, with a threshold of $0.05$ determining significance. In the trading phase, a rolling window is used to continuously recalculate the mean and standard deviation of the spread between the selected pairs. Trades are initiated based on the z-score of the spread, where a high z-score indicates a short position and a low z-score indicates a long position. Positions are managed and closed when the spread returns to a predefined z-score threshold. The strategy's performance is evaluated based on total return, annual percentage rate (APR), Sharpe ratio, maximum drawdown (maxDD), and maximum drawdown duration (maxDDD).

## B.8 EVALUATION METRICS

Following common conventions in the causal discovery literature, we evaluate our algorithms using the following metrics.

**Structural Hamming Distance (SHD).** Given two directed graphs $\mathcal{G}_1 = (V, E_1)$, $\mathcal{G}_2 = (V, E_2)$ over the same nodes with different sets of edges and let $A_1, A_2 \in \{0, 1\}^{n \times n}$ be their adjacency matrices. Then we define the structural hamming distance (SHD) as

$$\text{SHD} = \sum_{i,j \in [n]} |(A_1)_{ij} - (A_2)_{ij}|$$

**Normalized SHD.** In order to compare the recovery performance of our algorithms for different types of graph-sizes, the normalized SHD is defined as

$$\mathrm{SHD}_{\mathrm{norm}} = \frac{\mathrm{SHD}}{d(d-1)}$$

## B.9 COMPUTATIONAL COMPLEXITY

**Conditional Independence Test.** For KCIPT, we denote by $B$ the number of outer bootstraps, by $b$ the number of inner bootstraps, by $n$ the number of i.i.d. samples, and by $n_{MC}$ the number of Monte-Carlo samples. We spell out the computational complexity of all steps in performing a single conditional independence test.

- computing $B$ permutations of the n-samples: $\mathcal{O}(n)$
- the loop over the $B$ outer bootstrap contains in each iteration:
  - permuting matrices: $\approx \mathcal{O}(n^{2.4})$ (assuming fast matrix multiplication algorithms, otherwise $\mathcal{O}(n^3)$)
  - splitting matrices using indexing: $\mathcal{O}(n^2)$
  - computing RKHS distances: $\mathcal{O}(n^2)$
  - solving the linear program to find the best permutation (this depends on the solver, but for a primal-dual interior point methods it is $\mathcal{O}(\sqrt{n}\log(1/\epsilon)(mn + m^2))$, where $m$ is the number of constraints, $n$ the number of variables, and $\epsilon$ the accuracy tolerance for the solution. Overall, in our case with $m = n(n-1)$ constraints, we obtain: $\mathcal{O}(\sqrt{n}\log(1/\epsilon)(n^2(n-1) + n^2 \cdot (n-1)^2))$ which becomes $\mathcal{O}(n^{4.5}\log(1/\epsilon))$
  - permuting matrices: $\mathcal{O}(n^{2.4})$
  - computing the original statistic: $\mathcal{O}(n)$
  - loop over $b$ inner permutations
    * permuting matrices: $\mathcal{O}(n^{2.4})$
    * computing the permuted statistic: $\mathcal{O}(n^2)$
  - computing the p-value: $\mathcal{O}(b)$
- computing the original statistic: $\mathcal{O}(B)$
- for $n_{MC}$ Monte Carlo samples computing the null sample: $\mathcal{O}(B \cdot b)$
- computing the final p-value: $\mathcal{O}(n_{MC})$

**Causal discovery.** The computational complexity of our causal discovery algorithm is in the worst case $\mathcal{O}(d^{d_{\max}} \cdot d^2)$, where $d$ is the number of nodes and $d_{\max}$ is the maximal degree in the graph, meaning the maximal number of adjacencies of any node. For sparse graphs (e.g., graphs with a low maximal degree), the complexity can reduce to being polynomial in the number of nodes.

**Signature kernel.** The computational complexity of evaluating the signature kernel scales quadratically in the number of time points within a process using the `sigkerax` Python package with an RBF lifting kernel and again quadratically in the number of samples when computing the full Gram matrix. The is another package, `iisignature`, that allows for linear scaling in the number of time points when using a linear lifting kernel. Both implementation support highly parallelized execution on GPU accelerators.

## C BASELINES

In Section 4, we benchmark our method against other baselines, including Granger causality, CCM, PCMCI, and Laumann (Laumann et al., 2023). For the latter, we used their implementation provided in the `causal-fda` package. For the Granger-implementation for two variables ($d = 2$), we used Seabold & Perktold (2010), for CCM we used Javier (2021), and for PCMCI we used the `tigramite` package (Runge et al., 2019). In PCMCI, tests for edges are conducted by applying distance correlation-based independence tests (Székely et al., 2007) between the variables' residuals after regressing out other nodes using Gaussian processes. For SCOTCH implementation (Wang et al., 2024), we use the package `causica`. For SCOTCH, we always use a learning rate of $0.001$ and keep the same default parameters for the learning algorithm.

# D    DETAILS ON THE PARTIALLY OBSERVED SETTING

Since in the partially observed setting there could in principle be infinitely many unobserved variables, special tools are required. Usually, the graphical framework of *maximal ancestral graphs* (MAGs) (loosely speaking DAGs with also bidirected edges) is used, which encodes ancestral relations in the DAG $\mathcal{G}$ and aims at graphically representing conditional independencies implied by $\mathcal{G}$ involving only the (marginal of the) observed variables. The unique MAG $M^{\mathcal{G}}$ corresponding to the partially observed DAG $\mathcal{G}$ can be constructed via Zhang (2008) the following rules.

- For $v_1, v_2 \in V_{obs}$, $v_1 \,\text{---}\, v_2$ in $M^{\mathcal{G}}$ if and only if there exists an inducing path relative to $V_L$ between $v_1, v_2$ in $\mathcal{G}$.
- For each pair of adjacent vertices $v_1 \,\text{---}\, v_2$ in $M^{\mathcal{G}}$ we orient the edge between them as follows:
  - $v_1 \to v_2$ if $v_1 \in \text{an}_{v_2}^{\mathcal{G}}$ and $v_2 \notin \text{an}_{v_1}^{\mathcal{G}}$
  - $v_1 \leftarrow v_2$ if $v_2 \in \text{an}_{v_1}^{\mathcal{G}}$ and $v_1 \notin \text{an}_{v_2}^{\mathcal{G}}$
  - $v_1 \leftrightarrow v_2$ if $v_2 \notin \text{an}_{v_1}^{\mathcal{G}}$ and $v_1 \notin \text{an}_{v_2}^{\mathcal{G}}$

An *inducing path* is a path with all colliders on the path being in $\text{an}_{\{v_1,v_2\}}^{\mathcal{G}} \cap V_{obs}$ and all other nodes on the path in $V_L$. The obtained MAG $M^{\mathcal{G}}$ therefore preserves causal ancestral relations with respect to $\mathcal{G}$. The *Fast Causal Inference* (FCI) algorithm (Zhang, 2008) run on the observed marginal outputs an equivalence class of MAGs, known as a *Partial Ancestral Graph* (PAG), which captures the same adjacencies but leaves some endpoints unoriented as there are multiple MAGs with the same conditional independence relations. For example, in Figure 4 FCI cannot rule out an additional latent confounder between $A$ and $C$ such that the induced MAG has the same m-separation properties. Unlike FCI, by leveraging the direction of time, we can recover the full underlying MAG $M^{\mathcal{G}}$, which is substantially more informative than FCI's PAG. First, we establish adjacencies analogous to the first part of Algorithm 2 (or simply by running FCI with our symmetric criterion) and then follow the steps in the construction of the MAG from a DAG to orient these adjacent edges $v_i \,\text{---}\, v_j$, $v_i \neq v_j \in V_{obs}$:

- $v_i \to v_j$ if $X_0^i \not\perp\!\!\!\perp_{\text{sym}} X_{[0,T]}^j$ and $X_0^j \perp\!\!\!\perp_{\text{sym}} X_{[0,T]}^i$
- $v_i \leftarrow v_j$ if $X_0^i \perp\!\!\!\perp_{\text{sym}} X_{[0,T]}^j$ and $X_0^j \not\perp\!\!\!\perp_{\text{sym}} X_{[0,T]}^i$
- $v_i \leftrightarrow v_j$ if $X_0^i \not\perp\!\!\!\perp_{\text{sym}} X_{[0,T]}^j$ and $X_0^j \not\perp\!\!\!\perp_{\text{sym}} X_{[0,T]}^i$

We note that these criteria precisely encode the (non-)ancestral relationships in the MAG construction rules. Uniqueness of the obtained result follows from Richardson & Spirtes (2002, Corollary 3.10) as two ancestral graphs $\mathcal{G}_1, \mathcal{G}_2$ are equal, if they share the same adjacencies and ancestral relations.

We have thus constructed a sound and complete algorithm (assuming access to a CI oracle) to recover the unique MAG $M^{\mathcal{G}}$ for the ground truth DAG $\mathcal{G}$ in the partially observed SDE model. To illustrate our causal discovery algorithm in practice, we conduct experiments on an SDE model with a graph $\mathcal{G}$ as depicted in Figure 4 where $V_{obs} = \{A, B, C, D\}$, $V_L = \{U\}$, $E = \{(A,C), (B,C), (U,C), (U,D)\}$. While FCI run on data coming from a Markov model with this DAG is able to detect the adjacency structure, it is unable to decide whether $A$ is truly an ancestor of $C$. It will correctly infer the arrow-head into $C$, but cannot rule out an arrow-head at $A$, see Figure 4. Neural network approaches like SCOTCH that directly infer the functional relationships by model fitting are not able to deal with such partially observed settings and are typically expected to predict edges that do not exists in the ground truth graph, since they can not model potentially infinitely many unobserved processes. To demonstrate this concretely, we ran SCOTCH as well as our FCI-inspired algorithm on 100 SDEs with the adjacency from our example graph $\mathcal{G}$ and compare how often $A$ and $D$ are (falsely) adjacent in the output. SCOTCH predicted an edge between $A$ and $D$ in 88% and ours algorithm in only 8% of all settings, see also Figure 4.

