# OpenReview forum: "Signature Kernel Conditional Independence Tests in Causal Discovery for Stochastic Processes"
_ICLR.cc/2025/Conference — ICLR 2025 Spotlight_

### Official Review · Reviewer_4hZY · 2024-11-02

**Soundness:** 3
**Presentation:** 3
**Contribution:** 3
**Rating:** 8
**Confidence:** 4

**Summary:**

This paper introduces a novel approach for causal discovery in stochastic processes modeled by stochastic differential equations (SDEs). The authors propose a conditional independence (CI) test based on signature kernels, which enables the identification of causal relationships in continuous-time dynamical systems, even under irregular or partially observed data. Key contributions include a causal discovery algorithm that leverages the directionality of time, as well as an efficient signature kernel-based CI test.

**Strengths:**

1. The paper is well-motivated, presenting a clear need for improved causal discovery methods within the context of stochastic processes.
2. Extensive experimental evaluations support the effectiveness of the proposed approach. The method demonstrates consistent superiority over several baseline models, underscoring its potential contributions to the field.

**Weaknesses:**

1.	The paper’s claims regarding diffusion-dependence cases may be somewhat overstated. In particular, for SDE models involving "driving noise" (i.e., cases where the diffusion coefficient depends on $X_t$), certain **causal graphs may not be identifiable from observational data**. See example 5.5 in [1]. In such cases, identifying the generator of the SDE model—so as to identify the post-interventional distribution—may be a more reasonable goal, as explored in [2] for linear SDEs. This may also provide some insight into the **diffusion-dependence** results presented in Table 1, where the proposed method shows comparatively lower performance.

[1] Hansen, Niels, and Alexander Sokol. "Causal interpretation of stochastic differential equations." (2014): 1-24.

[2] Wang, Yuanyuan, et al. "Generator identification for linear SDEs with additive and multiplicative noise." Advances in Neural Information Processing Systems 36 (2024).

2.	The section on signature kernels in Section 2 is mathematically dense and may be challenging for readers unfamiliar with the topic. Including more intuitive explanations would enhance accessibility.
3.	Given the extensive use of mathematical notations, a summary table listing the key notations could be helpful for readers in following the manuscript’s developments.

**Questions:**

The experimental setup involves setting small parameters for self-loops (e.g.,  $a_{ii} \sim  U([-0.5, 0.5])$), while the parameters influencing causal effects between different variables are set higher (e.g., $a_{ij} \sim U([-2,-1] \cup [1, 2])$). While I understand this may help to emphasize inter-variable causal relationships, practical applications may not always exhibit this distinction so clearly. Additionally, could the authors clarify whether errors related to self-loops are reported in the Structural Hamming Distance (SHD) results?

---

> ### Author Response · Authors · 2024-11-15
> **Rebuttal by Authors**
>
> We thank the reviewer for their time and detailed constructive feedback. We address all mentioned concerns and questions one by one.
>
> ## Weaknesses
>
> 1. We apologize, this is on us. In the more detailed description of the SDE model in Appendix A.2 in the revised version and Appendix A.1 in the previous version (Intuition and details for the SDE model), we state in line 966 of the revised paper (line 920 in the previous version) that we assume the diffusion coefficient to be block-diagonal. This critical assumption – as rightfully pointed out by the reviewer – has disappeared from the main text (likely during our “shortening” process to satisfy the tight space constraints) and should have been clearly mentioned right in the “Data generating process” paragraph in line 50. We have added it back to where it belongs in the revised version (see also Appendix A.2 lines 967-971 in the revised version for the overall SDE structure).\
> The mentioned counterexample in [1] does not satisfy this assumption. With the block-diagonal diffusion structure (i.e., orthogonal diffusions) processes do have different distributions when given the same initial values (see also Theorem 4.1 and Assumption 2 (“orthogonal diffusion processes”) in [2] for this statement).
>
> [1] Niels Hansen and Alexander Sokol, “Causal interpretation of stochastic differential equations” \
> [2] Benjie Wang, Joel Jenning and Wenbo Gong, “Neural Structure Learning with Stochastic Differential Equations”
>
> 2. We agree. The space constraints made it difficult to prioritize what to provide intuition for while keeping the main text self-contained. We have extended Appendix A.4 (Signature Kernel Details) in the revised version (Appendix A.3 in the previous version) to include more intuition about signatures and the signature kernel:\
> In short, the signature transform of a (potentially multidimensional) continuous path (or time series) is an infinite sequence of iterated integrals, which form an expressive feature set for representing functions on paths, similarly to how vector monomials can be used to approximate functions on Euclidean spaces. For example, the depth 1 iterated integral describes the increment of each variable throughout time; depth 2 describes the signed area that is swept by the curve and chord connecting the beginning and end of it (an illustration of depth 1 and 2 of a path can be found in [1], see Figure 1); higher depths capture higher order geometric properties of the path such as volume and so on, but those are harder to visualize. We also added references [2, 3, 4] as introductory texts on the signature (kernel) method.
>
> [1] Morrill, James, Adeline Fermanian, Patrick Kidger, and Terry Lyons. "A Generalised Signature Method for Multivariate Time Series Feature Extraction." arXiv preprint arXiv:2006.00873 (2020).\
> [2] Chevyrev, Ilya, and Andrey Kormilitzin. "A primer on the signature method in machine learning." arXiv preprint arXiv:1603.03788 (2016).\
> [3] Lee, Darrick, and Harald Oberhauser. "The Signature Kernel." arXiv preprint arXiv:2305.04625 (2023).\
> [4] Cass, Thomas, and Cristopher Salvi. "Lecture notes on rough paths and applications to machine learning." arXiv preprint arXiv:2404.06583 (2024).
>
> 3. We thank the reviewer for this helpful suggestion. We have added a new section A.1 in the Appendix including a table with mathematical notations.

---

> > ### Author Response · Authors · 2024-11-15
> > **Rebuttal by Authors (part 2)**
> >
> > ## Questions
> >
> > Indeed, our key focus lies on inter-variable dependencies instead of on self-loops. We believe that in most practical settings, where one aims for causal discovery, (a) the key focus lies on inter-variable relationships, and (b) one may assume that self-loops always exist for essentially all variables. Assumption (b), that processes always feed into themselves is a reasonable assumption in physical systems and broadly applied, e.g., by PCMCI or as a blanket assumption in $\delta$-separation in [1].
> > The power analysis in Figure 3 indicates that for our (or any) test to have non-zero power in finding inter-variable relations, we require a certain “signal-to-noise” ratio (in our case the ratio of inter-variable strength to self-loop strength). Hence, as the reviewer stated, we chose the values for self-loops and inter-variable strengths accordingly.
> > One issue arising in evaluating self-loops as part of causal discovery is that particularly in graphs with sparse inter-variable dependencies, there may be more self-loops than inter-variable edges. Hence, the normalized SHD (where normalization by the overall number of possible edges is important for meaningful comparisons across graphs with different numbers of vertices) would be heavily “diluted” by the self-loops. Therefore, in our experiments we perform SHD for the inter-variable edges only. For example, in bivariate experiments we primarily aim at inferring $X \to Y$ (instead of the other direction or no edge at all), but self-loops would have twice the effect on the overall score than the edge of interest.
> >
> > [1] Didelez, Vanessa. "Asymmetric Separation for Local Independence Graphs." arXiv preprint arXiv:1206.6841 (2008).

---

> > > ### Comment · Reviewer_4hZY · 2024-11-20
> > > **Response to authors**
> > >
> > > I thank the authors for their thorough and detailed responses to my questions. As they have addressed all of my concerns, I am raising my score.

---

> > > > ### Author Response · Authors · 2024-11-21
> > > > **Response to reviewer**
> > > >
> > > > We are glad we could address the concerns and appreciate the updated score.

---

### Official Review · Reviewer_BsMU · 2024-11-02

**Soundness:** 4
**Presentation:** 4
**Contribution:** 3
**Rating:** 10
**Confidence:** 3

**Summary:**

The paper makes contributions to causal discovery in stochastic dynamical systems. In particular, it introduces a novel framework of conditional independence constraints in SDE processes, which are then leveraged to provide a causal discovery (CD) algorithm. The authors also provide a conditional independence (CI) test to evaluate these constraints from data. The authors evaluate the CD algorithm on synthetic data and compare it to other state-of-the-art baselines. They also evaluate their CI test on synthetic data and in a small case study on stock trading pairs.

**Strengths:**

The paper is well-written and a pleasure to read. All theoretical claims are backed by careful proofs, and care is taken to ensure reproducibility of their experimental results. The authors openly address the limitations of their methodology,  and additional interesting results and discussions are provided in a well-organized and comprehensive appendix.

I believe the paper earns its place in the suite of tools for causal inference with time-series data from dynamical systems. In my opinion, this is a high-quality paper that deserves acceptance.

**Weaknesses:**

The only major weakness is the lack of real-world experiments for the causal discovery algorithm, which I understand to be the main contribution of the paper. Only the CI test has a real-world data experiment on a downstream task, which I found creative and is well-documented in the appendix.

Naturally, because we are in causal inference, real-world data with a ground truth can be difficult to find. However, I would like to point the authors to two recent papers that provide real-world data with a causal ground truth and whose settings appear to be a good fit for the method in this paper:

**“[Causal discovery in a complex industrial system: A time series benchmark](https://arxiv.org/abs/2310.18654)” by Mogensen et al. (2023).**

The paper presents a real-world dataset from a dynamical system with partially observed data. The authors provide a ground-truth causal graph (section 2.4). The paper comes with a website (https://soerenwengel.github.io/essdata) with links to the dataset and preprocessing code.

**“[The Causal Chambers: Real Physical Systems as a Testbed for AI Methodology](https://arxiv.org/abs/2404.11341)” by Gamella et al. (2024)**

The authors build two physical devices, one of which (wind tunnel) produces real-world, time-series data from a dynamical system. There is a causal ground-truth graph for this system (Figure 3), which the authors use to benchmark the PCMCI+ algorithm (Figure 6a). I believe this is an extension of one of your baselines (PCMCI), which makes me suspect your method is also applicable. The authors provide a well-documented [notebook ](https://github.com/juangamella/causal-chamber-paper/blob/main/case_studies/causal_discovery_time.ipynb) to download the dataset and reproduce the PCMCI+ experiment. Using your method may be plug-and-play in this case.

There may be other suitable real-world datasets, but I found none after this search. A real-world experiment for the main contribution of the paper (the causal discovery algorithm) would further elevate the value of the paper, and I would be happy to raise my score as a result. This is only a suggestion, and my decision to accept is independent of whether the authors do this or not.

**Questions:**

Some minor typos and unclear sentences:

- Line 421: “even in the settings it was tailored to” -> what was tailored to these settings, SigKer or the state of the art?
- For figure 2, maybe explicitly say which graph is the lifted graph (right) and which is G (left)
- Line 157: “is inapplicable” -> “it is inapplicable”?
- Some pedantic styling comments:
- Line 105: you have double parenthesis with the citation to Laumaann et al.
- Lines 170,200,420:  you appear to be using hyphens (-) instead of em dashes (---) for interjections. See the JMLR formatting guide (under dashes): https://www.jmlr.org/format/format.html

---

> ### Author Response · Authors · 2024-11-15
> **Rebuttal by Authors**
>
> We thank the reviewer for their time, helpful feedback, and positive evaluation. We reply to the raised points one by one:
>
> ## Weaknesses
>
> **Real-world examples:** We thank the reviewer for these excellent suggestions.
> We have worked hard to benchmark our method on the proposed example from Gamella et al. 2024, which they curated in the notebook to also allow for comparison with an independent application of PCMCI+ (theirs). These experiments are currently running/work in progress and we are happy to provide results as soon as they come in.
>
> ## Questions
>
> Thanks for all these great catches, we have fixed/clarified all of these in the revised version.

---

> > ### Comment · Reviewer_BsMU · 2024-11-26
> >
> > Thank you for running the additional experiments. I'm very curious about the results! And thank you for addressing the minor comments.
> >
> > My decision to recommend acceptance still stands independently of the new results that the authors provide.

---

> ### Author Response · Authors · 2024-11-29
> **Real-world data**
>
> First, thanks again for pointing us to these relevant datasets. We started looking into the suggested wind- and light-tunnel experiments by Gamella et. al, 2024.
>
> Our initial thoughts were that:\
> (i) These measurements likely do not follow an SDE (e.g., externally prescribed input signals to actuators, various forms of delays, etc.), so we will operate under substantial model misspecification. \
> (ii) For each experiment, there is only a single trajectory available instead of multiple (stochastic) realizations as required in our setting. \
> (iii) In many experiments the externally controlled causes are correlated, yet depicted as independent.\
>
> Of course, some of these also affect PCMCI. The PCMCI approach in the notebook tackles (ii) by partitioning the single trajectory into multiple snippets of equal length and considers those to be different samples. We also adopt this approach for our method, so both PCMCI and our method operate under a violation of the iid assumption of sampled paths. This may in part explain the poor performance of PCMCI in the provided notebook. In summary, this is certainly a challenging setting.
>
> To ensure we can deliver useful results during the rebuttal period, we focused on the dataset from a “random walk windtunnel experiment” (wt_walks_v1) and considered a subgraph consisting of the main six variables—removing categorical and constant variables.
> We summarize the overall results of PCMCI (for several time lags tau) and our method (up to certain levels, i.e., sizes of the conditioning sets which grows in the outermost loop of our algorithm) in the following table:
>
> | Method        | τ    | Precision | Recall | SHD |
> |---------------|------|-----------|--------|-----|
> | PCMCI         | 1    | 0.43      | 0.33   | 10  |
> | PCMCI         | 10   | 0.33      | 0.22   | 11  |
> | PCMCI         | 20   | 0.50      | 0.33   | 9   |
> | ours (lvl 1)  | -    | 0.63      | 0.56   | 7   |
> | ours (full)   | -    | 0.60      | 0.33   | 8   |
>
> Given the model misspecification, non-iid samples, and the general difficulty of this real-world dataset–while far from perfect–our method performs relatively well and specifically manages to fairly reliably tease out which nodes are causes and which are effects. It appears as if (and this is the main reason we reported results at level 1) the fan measurements may be correlated (fans having to push against each other affect their speeds, drawn currents, speeds, effective resistance, etc). We suspect that when conditioning on one of them, we make the other independent of the sensors. Therefore, our method tends to cut too many edges too confidently when conditioning on other causes ultimately leading to a recall of only 33%.
>
> For further validation, we also ran the only existing time-series experiment in the light-tunnel “lt_walks_v1”/ “color_mix” (the “white_noise” experiment is not a time-series but a sequence of independent “static experiments”). In the light tunnel, the assumption of no interaction between externally controlled components is much more plausible to us. Again, leaving out constant variables (e.g. some additional photo-diodes with only discrete values {0,1,2} or polarization filter angles) we ran our method on the remaining graph of 9 nodes: brightness settings of the three main light-sources (red, blue and green LED) and the measurements of the light-intensity sensors for infrared and visible light. For this simplified setting, we obtain the following results:
>
> | Method        | τ    | Precision | Recall | SHD  |
> |---------------|------|-----------|--------|------|
> | PCMCI         | 1    | 0.50      | 0.33   | 18   |
> | PCMCI         | 10   | 0.73      | 0.44   | 13   |
> | PCMCI         | 20   | 0.70      | 0.39   | 14   |
> | ours (full)   | -    | 0.89      | 0.44   | 11   |
>
>
> Overall, our method can largely distinguish causal drivers from effects and gets their overall direction (into the effects) right. PCMCI sometimes fails at finding the correct direction “cause → effect”, but in our experiments (also searching over tau) performed better than depicted in the original jupyter notebook.

---

> > ### Comment · Reviewer_BsMU · 2024-12-02
> >
> > Thank you very much for running these experiments and all the work involved! I'm pleased by the positive results of your algorithm and your honest analysis of the limitations of the setup.
> >
> > I will need some time to dig into the setup of Gamella et al. to fully understand your results; in any case, I think the experiments offer an additional perspective for reasoning about your method and the underlying theory, which I find very interesting and I think rounds off the paper very nicely. I will update my score accordingly.
> >
> > Thank you for all your work!

---

> > > ### Author Response · Authors · 2024-12-03
> > >
> > > We appreciate your thoughtful feedback and are glad the experiments provide valuable insights. Thank you for your consideration in updating the score.

---

### Official Review · Reviewer_Fppw · 2024-11-03

**Soundness:** 2
**Presentation:** 3
**Contribution:** 3
**Rating:** 6
**Confidence:** 3

**Summary:**

The authors propose a sound and complete causal discovery algorithm for stochastic processes, incorporating a consistent signature kernel conditional independence test. The stochastic process is modeled using a stochastic differential equation (SDE), and the entire process is segmented into intervals that exhibit the Markov property with respect to an acyclic dependence graph.

**Strengths:**

1. The effect working on the 'continuous-time' path sequence is impressive.

2. The signature kernel conditional independence test is a novel CI test for path sequences.

3. The paper is well written, and the framework of the paper is straightforward.

4. The proposed algorithm has been applied to a series of simulations and one real-world dataset.

**Weaknesses:**

1. There seems to be no assumption sections. The acyclic assumption seems like a common assumption used in many causal discovery methods; however, it is more restricted in this paper as a cycle could be easily created if the causal relations exist for both $X^i_{0,s}$ to $X^j_{s,s+h}$ and $X^j_{s+h,s+2h}$ to $X^i_{s+2h,s+3h}$. Such pairs of causal relations are allowed in many causal discovery methods for time series, such as PCMCI. Therefore, though the authors claim that the proposed algorithm will not rely on the 'discrete-time' assumption, they did not discuss the impact of not assuming a time lag based on the 'discrete-time' assumption and the additional limitations from the acyclic assumption in this paper, compared to many previous causal discovery algorithms. Hence, the limitation discussion and comparison is not comprehensive. The acyclic assumption in this context is not very practical.

2. Does the proposed CI test only work in the specific setting of this paper? Could it be utilized outside of this setting, for more general time series?

3. In the limitations and requirements section, it is stated that the proposed algorithm far exceeds other existing causal discovery methods for time series data, which may not be the best way to frame the discussion. For instance, in part (b), it is mentioned that the proposed algorithm can handle confounders; however, there are already many algorithms that allow for confounders in time series, such as LPCMCI [1] and tsFCI [2]. Additionally, there are algorithms for non-stationary time series, such as CD-NOD [3], and those with special periodic patterns, such as PCMCI$_{\Omega}$ [4]. Therefore, it is difficult to conclude that the proposed algorithm far exceeds other related work, given different settings and assumptions.

4. The number of baselines in the experiment results is limited for both causal discovery and the CI test. Please refer to the questions section for more details.

5. Please correct me if I am mistaken, but it seems there is no computational complexity analysis or running time results provided.

6. I may have misunderstood, but does the first selected interval of $[0, s], [s, h]$ have to start at the beginning? Based on line 233, there are two copies of vertex $V$. Does this imply $h = T$? If not, how many intervals are possible? If multiple intervals are allowed, can the acyclic assumption be relaxed since different time-ordered intervals resemble the concept of "time lag" in a 'discrete-time' setting?

7. By intuition, is having $V_0$ and $V_1$ essentially a sub-sampling technique, where samples in $[0,s]$ and samples in $[s,h]$ are considered? The full causal graph that the algorithm aims to discover is restricted to this $V_0$ and $V_1$, and the estimated causal graph will be influenced by the intervals selected. Again, the full causal graph discussed here is different from the one in many causal discovery algorithms, referred to as the full time causal graph, which does not require sub-sampling and is more comprehensive. Therefore, this discussion needs to be handled with care.

[1]Gerhardus, Andreas, and Jakob Runge. "High-recall causal discovery for autocorrelated time series with latent confounders." Advances in Neural Information Processing Systems 33 (2020): 12615-12625.

[2]Entner, Doris, and Patrik O. Hoyer. "On causal discovery from time series data using FCI." Probabilistic graphical models 16 (2010).

[3]Huang, Biwei, et al. "Causal discovery from heterogeneous/nonstationary data." Journal of Machine Learning Research 21.89 (2020): 1-53.

[4]Gao, Shanyun, et al. "Causal discovery in semi-stationary time series." Advances in Neural Information Processing Systems 36 (2024).

**Questions:**

1. Could you briefly explain how to incorporate or partially incorporate PC and FCI into the proposed algorithm, given that both assume IID samples? Is any adjustment needed for non-IID samples?

2. Do KCIPT and SDCIT require IID samples as well? If so, a similar question arises as in item 1.

3. Are the simulated datasets used in the experiments 'continuous-time'? If so, how do you choose the discretization interval for PCMCI and other baselines that assume 'discrete-time'? Does using different discretization intervals influence their performance? How do you compare the output of PCMCI and the proposed algorithm, given that PCMCI includes time lags and may cause 'cycles' according to the definitions in this context?

4. Are you considering using more baselines designed for time series, particularly non-stationary time series? The number of baselines included is limited. Is there a specific reason for not using metrics such as F1 score, precision, and recall but just SHD?

5. Is there a power analysis for the conditional test, and how does it perform with different conditioning sets?

6. Could you explain how to obtain bi-directed edges as shown in Figure 4?

---

> ### Author Response · Authors · 2024-11-15
> **Rebuttal by Authors**
>
> We thank the reviewer for their time and detailed constructive feedback. We address all mentioned concerns and questions one by one.
>
> ## Weaknesses
>
> 1. This is a good point. On a purely technical level, the assumptions are described in the paragraphs “Data generating process” (line 50 onwards) with more details in Appendix A.2 of the revised version (Appendix A.1 of previous version) and “Limitations and requirements” (line 77 onwards). But we do not yet offer a detailed discussion of the acyclicity assumption in particular. \
> First, we absolutely agree with the reviewer and stand by our statement (lines 77-78) that “The key limitation of our setting is the assumptions [sic] of acyclicity”. As Example A.1 (Appendix A.3 in the revised version; Appendix A.2 in the previous version) shows, in the presence of cycles, it becomes impossible to infer the entire graph (even if there is no unobserved confounding). Hence, characterizing what can at best be learned in the cyclic case remains an exciting direction for future work. \
> In comparison to discrete-time methods, there is a fundamental distinction in the assumed data generating processes. Our SDE models considers a generative process as a differential equation yielding a continuous time evolution, where causal dependencies are encoded by which variables enter *the rates of change* of other variables. Instead, PCMCI and the corresponding “full time” graphs inherently assume a (typically fixed) autoregressive law on a fixed time grid mapping (multiple) previous time point(s) to the present as the true data generating mechanism. Causal dependence arises from variables at which previous times feed into the (static) function determining a given variable at the current time. \
> In our setting, we generally talk about $X^1$ causing $X^2$ without specifying specific times or time intervals since this causal dependence holds “infinitesimally”. The two settings and the corresponding acyclicity assumptions thus do not compare directly. Cycles in our setting would be of a different type than cycles arising “throughout time” in discrete-time settings. We further elaborate on these aspects in more details in our replies to Weaknesses 6+7 and to Question 3.
>
> 2. One of the key strengths of our CI test is that it also works beyond the SDE model, which we considered as a primary data generating mechanism. For example, Appendix B.6 (Conditional Independence Testing Beyond the SDE Model) shows that our test also works well on fractional Brownian motions (power analysis in Fig. 8) or in a “path dependent” functional data setting (lines 1834-1878 in the revised paper; 1714-1757 in the previous version).
> Table 10 (page 35) in the revised version in section B.6 (Table 9 in the previous version) shows results for a full causal discovery experiment on functional data. This specific experiment highlights that our approach performs well outside the SDE model, while the neural network based approach SCOTCH (explicitly tailored to the SDE model) cannot adapt to this different data generating process. Similarly, these results demonstrate how discrete-time approaches (PCMCI) fail when confronted with path-dependent interactions which they are not designed for. \
> More broadly, the signature transform (on which the signature kernel is built) can capture information about the filtration of stochastic processes [3], which is neglected when viewing them merely as path-valued random variables. This renders our CI test a powerful tool for many types of sequential data, including a variety of stochastic processes (SDEs only being a specific example).
>
> 3. We agree that our wording in line 83 is not precise enough–apologies! This statement was meant in the concrete context of “continuous time” approaches and we did not want to claim that our framework generalizes all existing methods along *all* dimensions (stationarity, acyclicity, partial observations, etc.), but is the first to simultaneously satisfy (a)-(d) in lines 83-90. We adjusted the wording to make this more clear. Specifically, our approach overcomes the “discrete-time” assumption, which all PCMCI-derivates (e.g. [1]) and [2] are subjected to (even though some of them deal with partial observations and/or non-stationarity). These require equidistant time steps and choosing “the right” sampling frequency (typically not known a-priori) as well as knowledge about the lag $\tau$. Hence, the reviewer is right that these approaches tackle different settings with different assumptions.
>
> 4. See replies to the questions (in particular Q4).

---

> > ### Author Response · Authors · 2024-11-15
> > **Rebuttal by Authors (part 2)**
> >
> > 5. Apologies for not having included the computational complexity of our method. We added these results in the revised version in a “Computational Complexity” section in the Appendix B.9 and refer to it in the main text. As there are multiple aspects to “computational complexity”, we provide computational complexity analyses separately for the permutation-based conditional independence test, the causal discovery algorithm, and computing the signature kernel.
> >
> > 6+7. In principle, the $s,h \in [0,T], s<t$ can be chosen arbitrarily. Specifically, the first interval does not have to start at $0$ – it just has to start before $s$ (the start of the second interval). More details on the choice of the two intervals can be found in Appendix B.1 “Kernel Heuristic and Interval Choice”. \
> > Our lifted dependence graph does not correspond to a “full time” or “unrolled” graph in discrete-time methods. We would like to go back a bit further to properly explain this distinction (see also reply to Question 3 below). The formal setting we consider accommodates continuous time paths, which is a critical distinction from methods like PCMCI (derivatives) that fundamentally assume a fixed time grid. Of course, in practice, all observed trajectories come as discrete observations. However, our setting merely assumes that these are actually discrete observations of an underlying continuous-time system. This naturally allows different coordinate processes to be measured at entirely different and arbitrarily spaced time points. For example $X^1$ could be observed at times $\{0, 0.01, 0.02, 0.5, 0.6, 0.9\}$, whereas $X^2$ is observed at times $\{0.1, 0.2, 0.25, 0.7, 0.95\}$. In our view, not allowing for different and/or irregular temporal observations blatantly neglects the fundamentally continuous nature of real-world systems. Most time series we measure are not generated in an autoregressive fashion with a fixed, built-in, natural time step. Therefore, we focus on differential equations, where the dynamics are determined infinitesimally leading to continuous evolutions. \
> > We argue that discrete-time methods for *causal discovery* like PCMCI (not arguing against  autoregressive methods for time series forecasting/classification etc) cannot consistently accommodate the view of a continuous underlying evolution. First, for the exemplary observation times of $X^1$ and $X^2$ above one cannot meaningfully speak about a “full time” graph, i.e., a graph with causal variables being observations at these specific times. One may then be tempted to interpolate both processes and re-sample them at a fixed discrete grid. Besides potentially requiring additional assumptions to choose an interpolation scheme, one would then have to choose “the right” time step and time lag (how many steps to look back into the past) – which also depends on the chosen time step – as the found graph critically depends on those two choices [1]. However, for a system that is not truly generated by a discrete autoregressive law but as a continuous evolution, there does not exist a “correct” (non infinitesimal) time step. Our causal relations arise from which variables enter *rates of change* of other variables, so we do not require a notion of time step or lag. \
> > This explains why it is not the case that we could map our setting onto a discrete-time setting by mapping different intervals onto “time points”, i.e., thinking of
> > $$
> > X_{[0,h]}, X_{[h, 2h]}, X_{[2h, 3h]}, \ldots, X_{[(N-1)h, Nh]}
> > $$
> > as discrete observations
> > $$
> > X_1, X_2, \ldots, X_N
> > $$
> > with time step $h$ and consider a causal graph between the individual segments $X_{[i\cdot h, (i+1)\cdot h]}$. Again, we do not require observations for all processes in all these intervals, whereas PCMCI cannot handle time points where some/all processes are not observed (not to be confused with partial observations, where entire processes are not observed). \
> > Instead, our time splitting simply serves the purpose of having one “past” segment and one “future” segment and we can split at an arbitrary point in time. In practice, this point may be chosen to retain similar numbers of samples in both segments. In the example time points above, something around 0.2 could be a reasonable choice. Critically, *because we assume stationarity, our resulting graph will not depend on the choice of this time point*. The causal relationship between any two consecutive intervals is the same across time and representative of what happens in the dependence graph (even though the exact functional form of the dependence may vary). There is no additional information to be gained about how the past is linked to the future from splitting the full interval into more than two segments.

---

> > > ### Author Response · Authors · 2024-11-15
> > > **Rebuttal by Authors (part 3)**
> > >
> > > ## Questions
> > >
> > > 1+2.  KCIPT and SDCIT do indeed require i.i.d. samples; thus, our causal discovery algorithms also require i.i.d. samples as input. In our setting, one solution path of all variables in $X$ is a single sample from the data generating process (SDE model). When drawing $n$ solution paths as independent realizations of a stochastic process (e.g., independent initial conditions and independent Brownian motions in the SDE), we obtain $n$ i.i.d. samples. Each such sample consists of multiple observations of the paths at different times, and these observations within a sample are not (and need not be) assumed to be i.i.d. In short, the observations within paths are not i.i.d., but different observed paths are i.i.d., which is all we need here. We have added this clarification in Appendix A.13 “I.i.d. Assumption of the Conditional Independence Test”.
> > > The logic of constraint-based causal discovery algorithms (PC, FCI, ours) is always the same: smartly but exhaustively test conditional independencies to remove or characterize edges in the graph. In that sense, our algorithm is in line with the mentioned examples. The essential parts for constraint-based causal discovery in new settings are to: (a) choose “the right kind of (conditional) independence to test in the given data modality” (l.222-228); (b) choosing “the right kind of separation criterion in the right graph” ($d$-separation in the lifted dependency graph); (c) link these two up via the data generating mechanism to satisfy a (global) Markov property (which we provide in Proposition 3.1 for our setting). Depending on the choices of (conditional) independence, construction of the considered graph, and choice of separation in the graph, one may obtain stronger or weaker results as output (or may not be able to establish the global Markov property). We’re happy to further elaborate on these points.
> > >
> > > 3. Yes, we simulate the synthetic datasets via SDE solvers. Concretely, we use the `diffeqsolve` function from the Python package `diffrax` with an Euler solver (such that the SDE solution converges to the Itô solution) and save the paths at a fine equidistant temporal grid from which we can then drop observations at random for irregular sampling (see, e.g., Figure 7 on how our test performs for high levels of data missingness). Since discrete-time methods like PCMCI fundamentally cannot handle irregular time observations, we run all methods on the full, dense, equidistant grid in all comparative experiments. \
> > > PCMCI variants are known to be sensitive to several assumptions [1], among them the critical and highly unrealistic assumption that not only all processes are observed at the same equidistant times, but also that the time interval is “just right.” Example 2 in [1] describes how one may obtain a completely wrong graph when the sampling interval is too large (subsampled from the hypothetical “just right” sampling frequency). Example 4 (time aggregation) and section IV.B in [1] provide further insightful details about the difficulties of satisfying the “discrete-time” related assumptions required for PCMCI. Hence, in terms of different dimensions of “performance”, subsampling will make the “discrete-time” baselines faster (simply because then fewer CI tests are required in the PCMCI case), but affects the validity of its output if the true “causal interaction time” (which for continuous systems is not necessarily a useful concept, see reply to Weaknesses 6+7 above) is not known a priori. Consequently, one may be tempted to use the highest possible sampling rate. However, it then becomes difficult to select the correct lag (how many timesteps to look back into the past), and larger lags affect the performance of PCMCI negatively as the conditioning sets grow with the lag.

---

> ### Author Response · Authors · 2024-11-15
> **Rebuttal by Authors (part 4)**
>
> 4. While there are many techniques for time series *forecasting*, not many methods exist for *causal discovery in continuous time* systems that allow for a sensible comparison in our setting. PCMCI variants are likely the most prominent and mohahast used for causal discovery in time series under the “discrete time” assumption, rendering them an important baseline to include. Similarly, Granger causality (with its variants such as CUTS [2]) and CCM are obligatory contenders due to their proven track record across domains. Laumann, focussing on functional data (but ignoring time) is a recent method built on similar ideas to our work and claimed SOTA on functional data, which we beat with our approach. Finally, SCOTCH (which appeared just earlier this year) is currently the strongest competitor considering a similar setting to ours (albeit not applicable in the presence of unobserved confounding, path-dependence, and tailored only to SDEs). With SCOTCH outperforming previous approaches based on similar techniques such as NGMs and Rhino, we focused on SCOTCH. We invested our limited computational resources into making sure to use the strongest version of (the computationally expensive) SCOTCH by exploring hyperparameter settings, instead of also running weaker baselines. \
> There is no particular reason we exclusively reported SHD in some experiments (instead of precision, recall, F1, etc) besides its common use in causal discovery and the fact that it is a convenient scalar metric for overall performance. In Appendix B.4 (Tables 6-8 in the revised version; previously Tables 5-7), we also report the performance of the CI test on a large suite of (conditional) independence tests, which form the “building blocks” of causal discovery (testing all kinds of (C)I in chain, fork, and collider structures). There, we report the “error,” which amounts to type I and II errors depending on whether the null hypothesis should or should not be rejected in the respective settings. Within the causal discovery context, these errors amount to falsely removing or falsely maintaining edges. We’re happy to also report precision, recall, and F1 if the reviewer finds this helpful?
>
> 5. A power analysis similar to Fig. 3 requires varying a single parameter with which we expect power to increase, i.e., something akin to a signal-to-noise ratio like the ratio “causal effect strength” to “self-loop strength” in Fig. 3. In conditional settings, the configuration of the three coordinate processes involves multiple interaction strengths and there is no natural choice for what to plot on the x-axis. Tables 6-8 (in the revised version; previously Tables 5-7) in Appendix B.4 (building blocks) report extensive results on the power of our CI test in different graph configurations (fork, collider, etc).
>
> 6. In the partially observed setting, bidirected edges between observed nodes are used to indicate situations like unobserved confounding, i.e., an unobserved variable is affecting both observed ones (e.g., X1 ← U → X2). Concretely, these can be found by the PC algorithm as follows: After running the PC-algorithm with the symmetric conditional independence criterion, we end up with the unoriented adjacencies of the “ancestral graph,” which only captures whether two coordinate processes can be separated by other observed variables or not (an edge/adjacency is then present/drawn). In the example in Fig. 7, this means we end up with A o—o C, B o—o C, C o—o D (a circle can be seen as placeholder for either arrowhead or tail). FCI would at this point use “orientation rules” (similar to the original PC algorithm) to at least correctly predict A o→ C resp. B o→ C meaning “A resp. B is not an ancestor of C” (which is the meaning of an arrowhead pointing towards C). Also leveraging time, we can use the unconditional independence test and check whether $X^A_0 \perp X^C_{[0,h]}$ holds and  $X^C_0 \perp X^A_{[0,h]}$ does not hold to orient A → C. Proceeding similarly for the edge C o—-o D and checking whether $X^C_0 \perp X^D_{[0,h]}$ does not hold resp. $X^D_0 \perp X^C_{[0,h]}$ does not hold, to orient the edge marks to obtain C ←→ D.
>
> [1] Runge, Jakob. "Causal network reconstruction from time series: From theoretical assumptions to practical estimation." Chaos: An Interdisciplinary Journal of Nonlinear Science 28.7 (2018). \
> [2] Cheng, Yuxiao, et al. "Cuts: Neural causal discovery from irregular time-series data." arXiv preprint arXiv:2302.07458 (2023). \
> [3] Salvi, Cristopher, et al. "Higher order kernel mean embeddings to capture filtrations of stochastic processes." Advances in Neural Information Processing Systems 34 (2021): 16635-16647.

---

> > ### Comment · Reviewer_Fppw · 2024-11-30
> >
> > Thank you for your detailed responses; I would like to increase the score.

---

> > > ### Author Response · Authors · 2024-12-03
> > >
> > > Thank you for your positive remarks. We're pleased our responses clarified your concerns and appreciate the score increase.

---

### Official Review · Reviewer_mqsA · 2024-11-04

**Soundness:** 4
**Presentation:** 4
**Contribution:** 4
**Rating:** 8
**Confidence:** 4

**Summary:**

This paper introduces an innovative approach to uncovering causal relationships in stochastic processes, using conditional independence (CI) tests based on signature kernels to detect causal links within stochastic differential equation (SDE) models. It presents a comprehensive algorithm to reconstruct causal graphs, even with partially observed data and irregular sampling patterns. Benchmark tests show this method outperforms existing causal discovery techniques (in small graphs) in continuous-time settings, showing particular strength in cases with incomplete data and path-dependence (without need for hyper parameter tuning)

**Strengths:**

- Provides a solid framework for causal inference in continuous-time SDEs, going beyond the limitations of traditional discrete-time models.
- Introduces a practical CI test using signature kernels, suited for handling path-dependent random variables.
- Shows strong performance with incomplete data and irregularly sampled time series.
- Empirical tests and real-world examples, like pairs trading, demonstrate the algorithm’s practical effectiveness and accuracy.
- Method is not heavily dependent on hyper parameters.

**Weaknesses:**

- The paper assumes stationarity and acyclicity, which may restrict its use in scenarios where causal relationships change over time.
- Limited discussion on performance of the algorithm if the IC Oracle is wrong.
- While I appreciate the rigor in the paper, the length of the paper, considering the appendix is more than $2 /3$ of the page limit. Might be more suitable for a journal setting.

**Questions:**

- The  Oracle used for this algorithm, how likely it is to be wrong? How much it would affect the performance?
- What happens if the underlying causal graph has a cycle? How does the algorithm handle this situation?
- How does the algorithm handles the case where the observation in time is limited (subsampled) ?

---

> ### Author Response · Authors · 2024-11-15
> **Rebuttal by Authors**
>
> We thank the reviewer for their time, helpful feedback, and positive evaluation. We reply to the questions one by one:
>
> * “what if the oracle is wrong?”: While the oracle is never wrong (by definition), any practical CI test will sometimes make mistakes on finite data (type I and II errors). All constraint-based causal discovery methods (including PC, FCI, and all variants of PCMCI in the time series setting) perform adaptive (C)I tests, where subsequent tests depend on the outcomes of previous tests. While this is a widely acknowledged challenge faced by all constraint-based methods, how the final graph depends on type I and II errors remains an important open problem in the field. Spirtes and Meek already discussed it in 1995 [1]. They added a (from today’s viewpoint somewhat limited) simulation study showing how sensitive different methods are to the different types of errors without a clear-cut conclusion (different benefits and drawbacks at different sample sizes for “adjacency” detection and “arrowhead” detection, respectively). Instead, [2] provides theoretical results showing that the PC algorithm is uniformly consistent for high-dimensional settings under a mild sparsity assumption on the DAGs, where the number of nodes can grow quickly with the sample size. However, they critically rely on the restrictive assumption that the observational distribution is a multi-variate Gaussian for their finite sample results, where a strong theory for testing CI with partial correlations is available. \
> In summary, the reviewer puts their finger on an important open problem in constraint-based causal structure learning, highlighting the need for empirical simulation studies to evaluate the performance of any concrete method (as provided in our paper). We added this discussion with the two references as a novel Appendix section A.12 and linked it to the importance of having a consistent CI test (Appendix A.14 in the revised version; previously Appendix A.11).
> * “what if there are cycles?” As we show in Example A.1 (Appendix A.3 in the revised version; previously Appendix A.2), in the presence of cycles, it becomes impossible to infer the entire graph (even if there is no unobserved confounding). On cyclic graphs, our Algorithm 1 will output a supergraph of the true graph (in the oracle setting), i.e., if an edge is in the true graph, it is also in the algorithm's output, guaranteeing edges will not falsely be removed. The output may include additional edges that are not in the true graph.\
> In such situations where the unique true graph is impossible to obtain, one would typically attempt to characterize the entire set of graphs compatible with the testable conditional independencies (Markov equivalence class) in a parsimonious fashion (similar to the CPDAG for Markov equivalence classes of DAGs). A comprehensive characterization of the equivalence class of graphs (entailing the same set of separations) in the cyclic setting is an exciting direction for future work, which we touch upon in Section 5.
> * “subsamples”: We are not 100% sure we understand what is meant, but assume this question aims at the algorithm's sensitivity to fewer and fewer observed time points in the individual paths? Appendix B.3 (Figure 7) provides an analysis of how the CI test power (the critical component here) changes with the level of “data missingness,” i.e., when dropping more and more observation points in the paths. Test power remains surprisingly strong, even for high levels of data missingness. (The influence of the number of paths, i.e., the sample size, is assessed in Figure 3 and varied throughout many of the other experiments.)
>
>
> [1] Spirtes, Peter, and Christopher Meek. "Learning Bayesian networks with discrete variables from data." KDD. Vol. 1. 1995.\
> [2] Kalisch, Markus, and Peter Bühlman. "Estimating high-dimensional directed acyclic graphs with the PC-algorithm." Journal of Machine Learning Research 8.3 (2007).

---

> > ### Comment · Reviewer_mqsA · 2024-11-27
> > **thank you for clarification.**
> >
> > I want to thank the authors for their clarifications. My decision remains accept.

---

### Meta-Review · Area_Chair_zg7E · 2024-12-23

**Metareview:**

This paper tackles the challenging and important problem in causal discovery of extracting causal relationships from temporal data governed by a system of stochastic differential equations (SDEs). The authors show how to extract a causal graph from a prototypical system, and propose methods for learning this graph. Reviewers were unanimously in favour of acceptance, with 3/4 reviewers scoring the paper >=8. I recommend the paper be accepted as an oral presentation given the novelty and significance of its contributions.

**Additional Comments On Reviewer Discussion:**

The paper was initially received positively, and after discussion, scores were increased across the board, reflecting reviewers' interest and support for the proposed approach.

---

### Decision · Program_Chairs · 2025-01-22

Accept (Spotlight)